# Biomass burning aerosol over the Amazon: analysis of aircraft, surface and satellite observations using a global aerosol model

Carly L. Reddington[1], William T. Morgan[2], Eoghan Darbyshire[2], Joel Brito[3,4], Hugh Coe[2], Paulo Artaxo[3], Catherine E. Scott[1], John Marsham[1], Dominick V. Spracklen[1]

[1] School of Earth and Environment, University of Leeds, Leeds, UK.
[2] Centre of Atmospheric Sciences, School of Earth and Environmental Science, University of Manchester, Manchester, UK.
[3] Physics Institute, University of São Paulo, São Paulo, Brazil.
[4] Now at: Laboratoire de Météorologie Physique, Université Clermont Auvergne, Aubière, France.

*Correspondence to*: C. L. Reddington (c.l.s.reddington@leeds.ac.uk)

**Abstract.** Vegetation fires emit large quantities of aerosol to the atmosphere impacting regional air quality and climate. Previous work has used comparisons of simulated and observed aerosol optical depth (AOD) in regions heavily impacted by fires to suggest emissions of aerosol particles from fires may be underestimated by a factor of 2-5. Here we use surface, aircraft and satellite observations made over the Amazon during September 2012 along with a global aerosol model to improve understanding of aerosol emissions from vegetation fires. We apply three different satellite-derived fire emission datasets (FINN, GFED, GFAS) in the model. Daily mean aerosol emissions in these datasets vary by up to a factor 3.7 over the Amazon during this period, highlighting the considerable uncertainty in emissions. We find variable agreement between the model and observed aerosol mass concentrations. The model well reproduces observed aerosol concentrations over deforestation fires in the western Amazon during dry season conditions with FINN or GFED emissions and during dry-wet transition season conditions with GFAS emissions. In contrast, the model underestimates aerosol concentrations over savannah fires in the Cerrado environment east of the Amazon Basin with all three fire emission datasets. The model generally underestimates AOD compared to satellite and ground stations, even when the model reproduces the observed vertical profile of aerosol mass concentration. We suggest it is likely caused by uncertainties in the calculation of AOD, which are as large as ~90%, with the largest sensitivities due to uncertainties in water uptake and relative humidity. Overall, we do not find evidence that particulate emissions from fires are systematically underestimated in the Amazon region and we caution against using comparison with AOD to constrain particulate emissions from fires.

## 1 Introduction

Vegetation and peat fires (open biomass burning) are a major source of particulate matter (aerosol) to the atmosphere (van der Werf et al., 2010, Langmann et al., 2009) dominating the aerosol burden in many tropical regions (Lelieveld et al., 2015). There is considerable uncertainty in the magnitude of aerosol emissions from tropical fires (Reddington et al., 2016), hindering estimates of the impacts of fire on weather (Kolusu et al., 2015; Archer-Nicholls et al., 2016), climate (Rap et al.,

2013; Thornhill et al., 2018) and human health (Johnston et al., 2012, Marlier et al., 2013, Reddington et al., 2015, Reid et al., 2016). Here we evaluate the Global Model of Aerosol Processes (GLOMAP; Spracklen et al., 2005) against a comprehensive set of measurement data (including surface, aircraft and satellite observations) collected during the South American Biomass Burning Analysis (SAMBBA) field campaign in September/October 2012 over the Amazon basin. Our aims are to: 1) quantify the effects of biomass burning emissions on the aerosol distribution over the Amazon; and 2) explore how different fire emissions datasets affect simulated aerosol concentrations over this region.

Models systematically underestimate aerosol optical depth (AOD) in regions impacted by tropical biomass burning, potentially suggesting that emission datasets underestimate aerosol emissions (Reddington et al., 2016). Fire emissions datasets are typically created through combining information on fire location and extent from satellite remote sensing with estimates of biomass consumption and species-specific emission factors (Langmann et al., 2009). Emissions could be underestimated due to missing fire detections or uncertainties in burned area (Randerson et al., 2012), fuel consumption (van Leeuwen et al., 2014; Andela et al., 2016), or emission factors (van Leeuwen et al., 2013; Stockwell et al., 2016). Agreement between bottom-up and top-down approaches for carbon emissions from fires is typically better than for particulate matter (PM) (Yin et al., 2016), suggesting that uncertainties in burned area or fuel loads do not dominate.

Estimating emissions of PM from fires is further complicated by the emission of a range of semi-volatile and intermediate volatility organic compounds that can contribute to aerosol formation (Grieshop et al., 2009; Jathar et al., 2014). These processes are poorly understood and are not treated in many models. Observational studies report varying amounts of secondary organic aerosol (SOA) formation in different biomass burning plumes. Analysis of Siberian biomass burning plumes (Konovalov et al., 2017) and southern African savannah and grassland fire plumes (Vakkari et al., 2018) show substantial in-plume SOA formation, whereas other studies report little SOA formation in tropical biomass burning plumes (Jolleys et al., 2012). At the global scale, a recent modelling study (Tsimpidi et al., 2016) estimates that 30% of organic aerosol (OA) in biomass burning aerosol originates from direct particulate emissions with the remainder being formed in the atmosphere. Analysis of OA:CO ratios in biomass burning plumes during the SAMBBA campaign suggests limited SOA formation from Amazon fires (Brito et al., 2014).

Top-down studies typically use AOD, available from satellite remote sensing, to help constrain aerosol emissions from fires. In addition to particle mass concentration, simulated AOD is sensitive to assumptions about particle size, chemical composition, vertical profile of aerosol, optical properties, water uptake as well as meteorology and model resolution (Brock et al., 2016). Reddington et al. (2016) found that a global aerosol model showed better agreement with observed PM mass concentration compared to AOD, potentially suggesting that some of the discrepancy between top-down and bottom up studies may be connected to the calculation of AOD.

To help explore these issues we analyse observations from the SAMBBA field campaign over the southern Amazon during the end of the dry season and transition to wet season. The Amazon exhibits a very strong seasonal cycle in aerosol concentrations (Martin et al., 2010). In the wet season (~December to ~May), PM2.5 (particulate matter with diameters

smaller than 2.5 µm) concentrations in the central Amazon are ~1.5 µg m$^{-3}$ and aerosol number concentrations of 220 cm$^{-3}$ (Poschl et al., 2010, Artaxo et al., 2013), some of the lowest concentrations observed in a terrestrial environment. In the dry season (~June to ~November), fires occur across Southern Amazonia, resulting in aerosol concentrations that are an order of magnitude higher (PM2.5 concentrations of >30 µg m$^{-3}$ and aerosol number concentrations > 20 000 cm$^{-3}$ (Artaxo et al.,
2013).

Fires in the Amazon are a consequence of both climate and human activity (van Marle et al., 2017). There was relatively little fire activity in the Amazon before the mid-1980s (van Marle et al., 2017), when large scale clearance of the Amazon forests began. Fire is used to clear forest and vegetation resulting in positive relationships between the rate of deforestation and fire activity in the Amazon (Aragao et al., 2008, Reddington et al., 2015, van Marle et al., 2017). A reduction in the rate
of deforestation across the Brazilian Amazon between 2002 and 2012 (Hansen et al., 2013) has led to reductions in deforestation-related fires (Reddington et al., 2015) and observed reductions in CO (Jiang et al., 2017) and AOD (Reddington et al., 2015). Fires are also used to maintain agricultural and pastoral land and may escape into surrounding forest leading to forest degradation (Chen et al., 2013a) and resulting in a disconnection between fire and deforestation (Aragao and Shimabukuro, 2010, Cano-Crespo et al., 2015). There has been reduction in area burned by fires in SW Amazon
and increase in area burned further east during the last decade (Andela et al., 2017). Droughts enhance the occurrence of fire (Chen et al., 2013b) with seasons of increased large fire occurrence coinciding with the Amazon droughts of 2005, 2007 and 2010 (Chen et al., 2013a).

Aerosol from fires degrades air quality with negative impacts on human health (Marlier et al., 2013, Reddington et al., 2015, Koplitz et al., 2016, Crippa et al., 2016). Inhalation of smoke from fires in the Amazon causes DNA damage and death of
human lung cells (de Oliveira Alves et al., 2017), impacts lung function (Jacobson et al., 2014), causes increased hospitalisations for respiratory diseases (Smith et al., 2014) and is estimated to result in thousands of mortalities each year (Reddington et al., 2015). Estimates on the health impacts of degraded air quality from fires require accurate information on the magnitude of particulate emissions from fire. A range of policy interventions will be necessary to reduce Amazonian fire (Morello et al., 2017).

Here we combine detailed observations of aerosol vertical profiles made over the Brazilian Amazon during the dry season of 2012 with surface observations, remote sensing and an aerosol model to better understand model representations of the magnitude and spatial distributions of particulate emissions from biomass burning.

**2. Method**

**2.1 GLOMAP global aerosol model**

We used the TOMCAT chemical transport model (Chipperfield, 2006) coupled to the GLOMAP global aerosol model (Spracklen et al., 2005) to simulate aerosol during the SAMBBA campaign. Below we describe the features of the model

relevant for this study, please see Spracklen et al. (2005) and Mann et al. (2010) for more detailed descriptions of the model and see Reddington et al. (2016) for further details of the model set-up used here.

Large-scale atmospheric transport and meteorology in TOMCAT are specified from European Centre for Medium-Range Weather Forecasting (ECMWF) ERA-Interim reanalyses, updated every six hours and linearly interpolated onto the model time step. The model has a horizontal resolution of 2.8°×2.8° with 31 vertical model levels between the surface and 10 hPa. The vertical resolution in the boundary layer ranges from ~60 m near the surface to ~400 m at ~2 km above the surface.

The aerosol size distribution is represented by a two-moment modal aerosol scheme (Mann et al., 2010). GLOMAP includes black carbon (BC), particulate organic matter (POM), sulfate ($SO_4$), sea spray and mineral dust. Concentrations of oxidants are specified using monthly mean 3-D fields at 6-hourly intervals from a TOMCAT simulation with detailed tropospheric chemistry (Arnold et al., 2005) linearly interpolated onto the model time step.

Wet removal of aerosol in GLOMAP occurs by two processes: 1) in-cloud nucleation scavenging, calculated for both large-scale and convective-scale precipitation based on rain-rates diagnosed from successive ECMWF ERA-Interim reanalysis fields; and 2) below-cloud impaction scavenging via collection by falling raindrops. For dry deposition of aerosol, GLOMAP calculates the wind speed and size-dependent deposition velocity due to Brownian diffusion, impaction and interception. Detailed descriptions of the dry and wet aerosol removal process are in Mann et al. (2010).

Anthropogenic emissions of sulfur dioxide ($SO_2$), BC and organic carbon (OC) were specified using the MACCity emissions inventory for 2010 (Lamarque et al., 2010). Open biomass burning emissions of $SO_2$, BC and OC are described in Sect. 2.2. Primary carbonaceous aerosol particles are assumed to be non-volatile and are emitted into the model with a fixed log-normal size distribution, assuming a number median diameter of 150 nm for biomass burning emissions and 60 nm for fossil fuel emissions and modal width (σ) of 1.59. We convert primary OC to POM using a prescribed POM:OC ratio of 1.4, which is at the lower end of the range prescribed in other global models (1.4 to 2.6) (Tsigaridis et al., 2014). Monthly mean emissions of biogenic monoterpenes are taken from the Global Emissions InitiAtive (GEIA) database (Guenther et al., 1995). Monoterpenes are oxidised to form a product that condenses irreversibly in the particle phase (Scott et al., 2014). Size-resolved emissions of mineral dust are prescribed from daily varying emissions fluxes provided for AEROCOM (Dentener et al., 2006).

### 2.1.1 Description of model simulations

We performed four main model simulations with GLOMAP: one simulation excluding open biomass burning emissions ("noBBA"), and three simulations including open biomass burning emissions (using three different open biomass burning emissions datasets: "FINN", "GFED" and "GFAS"; see Sect 2.2). Simulations were run from 1st January 2003 to 31st December 2012, using ECMWF ERA-Interim reanalyses that correspond to the simulation date/time. The model aerosol fields were generated from an initially aerosol-free atmosphere initialised on 1st October 2002 and spun-up for 92 days to produce a realistic aerosol distribution (Spracklen et al., 2005). The model was set up to output 3-D monthly-mean global

fields and 1-D daily-mean vertical profiles at the locations of ground and AERONET stations (Sect. 2.3). In this study, we show results from the model simulations for the year 2012. Specifically for this study, the model was set up to output additional 3-D fields every hour between July and November 2012 over a South American domain.

### 2.1.2 Calculation of aerosol optical depth

5   AOD was calculated from the simulated aerosol size distribution as in Reddington et al. (2016), using Mie theory assuming spherical particles (Grainger et al., 2004) that are internally mixed within each log-normal mode. Modelled AOD was calculated at specific wavelengths to match observations (500 nm and 550 nm), using component-specific refractive indices at the closest wavelength available from Bellouin et al. (2011).

The aerosol hygroscopicity in the AOD calculation was obtained directly from GLOMAP using the aerosol water uptake 10   calculated online in the model using Zdanovskii-Stokes-Robinson (ZSR) theory (Stokes and Robinson, 1966) (described in Sect. S1.1). We explore the sensitivity of simulated AOD to the calculation of aerosol water uptake in Sect. 3.5, by also using the κ-Köhler scheme (Petters and Kreidenweis, 2007) to calculate an offline estimate of water uptake (described in Sect. S1.2). The ZSR and κ-Köhler methods used in this study represent high and low aerosol water uptake cases, respectively (Reddington et al., 2016). In Sect. 3.5 we also explore the sensitivity of simulated AOD to assumed refractive 15   indices and aerosol mixing state.

### 2.2 Biomass burning emissions

### 2.2.1 Biomass burning emissions in GLOMAP

We used three different emissions datasets of aerosol from open biomass burning: the National Centre for Atmospheric Research Fire Inventory (FINN) (Wiedinmyer et al., 2011), the Global Fire Emissions Dataset (GFED) (van der Werf et al., 20   2010) and the Global Fire Assimilation System (GFAS) (Kaiser et al., 2012). We use daily mean fire emissions from FINN version 1.5, GFED version 4.1s (Mu et al., 2011; van der Werf et al., 2017) and GFAS version 1.2 (hereafter referred to as GFAS, FINN and GFED respectively).

Brito et al. (2014) analysed OA:CO ratios and found little enhancement of OA in fire plumes during the SAMBBA campaign, suggesting that SOA formation is limited or balanced by loss of OA through volatilization. A recent study 25   analysed airborne in-situ observations of biomass burning carbonaceous aerosol during SAMBBA and found find limited evidence for net increases in aerosol mass through atmospheric aging (Morgan et al., 2019). These observations suggest that SOA formation in plumes may be occurring on short timescales (Morgan et al., 2019) but since a net increase in OA mass was not observed in the regional-scale analyses of Brito et al. (2014) and Morgan et al. (2019), we do not include any SOA formation associated with biomass burning emissions.

30   Fires can inject emissions above the surface due the buoyancy of the fire plume. Marenco et al. (2016) analysed Lidar data during the SAMBBA campaign and found that the mean height of aerosol layers was 2.0±0.4 km, suggesting that the

majority of the aerosol is injected into the boundary layer. Fire emissions in GLOMAP are distributed vertically over six ecosystem-dependent altitudes between the surface and 6 km according to Dentener et al. (2006). Over Brazil ~53% of emissions were injected below 500 m elevation, ~30% between 500 m and 1000 m elevation, and ~17% between 1000 m and 3000 m elevation. We also performed a sensitivity simulation where fire emissions were injected into the model surface layer. We evaluate the vertical profile of simulated aerosol in Sect. 3.2.

## 2.2.2 Description of biomass burning emissions datasets

FINN, GFED and GFAS provide daily fire emissions of aerosol and gas-phase species. FINN emissions are available from 2002 to 2018 on a 1 km$^2$ grid (Wiedinmyer et al., 2011; available at http://bai.acom.ucar.edu/Data/fire/); GFAS emissions are available from 2003 to present on a $0.1^0$x$0.1^0$ grid (Kaiser et al., 2012; available at: https://apps.ecmwf.int/datasets/data/cams-gfas/); GFED emissions are available from 2003 to 2018 (monthly emissions are available from 1997) on a $0.25^0$x$0.25^0$ grid (van der Werf et al., 2017; available at: https://www.geo.vu.nl/~gwerf/GFED/GFED4/). Reddington et al. (2016) include a detailed a description of the FINN version 1.0, GFED version 3 and GFAS version 1.0 datasets. A brief description of the updated datasets is given below.

FINN emissions are based on the location and timing of active fire detections from the MODIS Fire and Thermal Anomalies Product (Giglio et al., 2003); using MODIS Land Cover Type and Vegetation Continuous Fields products to specify land cover classes and identify fractions of tree and non-tree vegetation, and bare ground. A burned area is assigned to each fire count (0.75 km$^2$ fires detected on grassland and savannah land cover classes and 1 km$^2$ for all other fire detections), with adjustments made to the assumed burned area if the fire pixel extends partially over bare ground. Estimates of biomass loading are taken from Hoelzemann et al. (2004) and emission factors for each species are taken from Akagi et al. (2011). The version 1.5 dataset used here (acquired in 2014) includes emission factors updated to incorporate measurements published in Yokelson et al. (2013) and Akagi et al. (2013) (for more information see updates described here: http://bai.acom.ucar.edu/Data/fire/).

GFAS uses the observed geo-location of active fires (like FINN) combined with fire radiative power (FRP) derived from the MODIS instrument. The FRP fields are corrected for observation gaps due to partial cloud-cover and/or spurious signals (e.g., from volcanoes, gas flares etc.). FRP is converted to the combustion rate of dry matter using land-cover-specific conversion factors based on data from GFED (Heil et al., 2010; Kaiser et al., 2012). Trace gas and aerosol emission rates are calculated using updated emission factors based on Andreae and Merlet (2001). The version 1.2 dataset is at higher spatial resolution than previous versions with improvements made to the processing and assimilation of satellite observations.

GFED emissions are based on estimates of burned area (Giglio et al., 2013), active fire detections, and plant productivity from the MODIS instrument. To derive total carbon emissions the satellite datasets are combined with estimates of fuel loads and combustion completeness for each monthly time step from the Carnegie-Ames-Stanford-Approach biogeochemical model. Carbon emission fluxes are converted to trace gas and aerosol emissions using species specific emission factors based

on Akagi et al. (2011) and Andreae and Merlet (2001) with updates provided in 2013 by M.O. Andreae. Significant updates in the version 4.1s dataset relevant for this study include: i) higher spatial resolution; ii) new burned area estimates (Giglio et al., 2013) with additional contributions from small fires (Randerson et al., 2012); and iii) improved representation of fuel consumption (see van der Werf et al., (2017) for more detail on updates from version 3 to version 4).

## 2.2.3 Comparison of biomass burning emissions datasets

Figure 1 compares total annual OC emissions from GFAS, FINN and GFED. The figure shows long-term (2002-2012) mean annual total emissions as well as annual total emissions in 2012, the year of the SAMBBA field campaign (total emissions for the SAMBBA field campaign are shown in Fig. S1). For the long-term mean, all datasets show broadly similar spatial patterns with greatest OC emissions across the arc-of-deforestation (roughly 65-50°W, 8-14°S). Total annual BC emissions show very similar spatial patterns to the OC emissions shown in Figs. 1 and S1. Table 1 compares total annual BC and OC emissions from the three datasets and Figs. S2, S3 and S4 compare the total daily OC emissions for the 2012 dry season and the SAMBBA campaign period.

For annual total emissions averaged over the 2002-2012 period, FINN emissions are greater than GFED and GFAS across regions of deforestation in the western Amazon, but lower than GFED and GFAS in eastern Amazonia (50-40°W, 4-15°S). Annual total aerosol (OC+BC) emissions averaged over 2002-2012 differ by up to a factor 2.4 (FINN:GFAS) in the western region and by up to a factor 1.9 (GFAS:FINN) in the eastern region. Matching 2002-2012 comparisons, FINN emissions in 2012 (and during the SAMBBA period) were also greater than GFED and GFAS over deforestation regions of western Amazonia and lower than GFED and GFAS in eastern Amazonia. Annual total OC+BC emissions in 2012 vary by up to a factor 3.5 in the west (FINN:GFAS) and up to a factor 2.1 in the east (GFAS:FINN). Pereira et al. (2016) also reported that FINN had lower (higher) aerosol emissions in the eastern (western) Amazon compared to GFAS during the SAMBBA period. Reddington et al. (2016) reported similar patterns for comparison of GFAS version 1.0, FINN version 1.0 and GFED version 3 emissions. During the SAMBBA campaign, daily mean aerosol (OC+BC) emissions differ between the different datasets by up to a factor 3.7 in the western Amazon (FINN:GFAS) and by up to a factor 2.4 in the eastern Amazon (GFAS:FINN).

Figure 1 also shows the difference in annual total OC emissions between 2012 and the 2002-2012 mean (very similar spatial patterns are seen for BC emissions). All three datasets show consistently lower emissions in 2012 compared to 2002-2012 across the arc-of deforestation in western Brazil and Bolivia (70°-50°W, 8°-18°S). OC emissions in 2012 in the western Amazon were 4-47% lower than the 2002-2012 mean (Table 1). Aerosol emissions from fire in Brazil have declined over this period, related to reductions in deforestation (Reddington et al., 2015) and consistent with observed declines in CO (Jiang et al., 2017) and AOD (Reddington et al., 2015). Figure S5 shows a reduction in the area dominated by deforestation type fires (and an increasing dominance of savannah-type fires) in 2012 relative to the 2002-2012 mean. In 2012, emissions were greater than the 2002-2012 average across much of Peru, possibly due to increased deforestation there (Kalamandeen et

al., 2018). In the eastern Amazon, emissions in 2012 were 30-96% greater than the 2002-2012 mean (Table 1), with largest differences in GFAS and GFED datasets.

## 2.3 South American Biomass Burning Analysis (SAMBBA)

We used observations from the South American Biomass Burning Analysis (SAMBBA) campaign. Aircraft and ground observations took place from 13 September to 3 October 2012. We separate the campaign into the dry season (Phase 1; 13 to 22 September) and the dry-wet season transition (Phase 2; 23 September to 3 October) following Brito et al. (2014). Figure 2 shows locations of aircraft flights and surface measurement sites.

### 2.3.1 Aircraft observations

The BAe-146 research aircraft from the Facility for Airborne Atmospheric Measurements (FAAM) made 20 research flights with measurements of a range of gas-phase and aerosol species. We use measurements of OA and sulfate mass in the 50 - 750 nm size range from an Aerosol Mass Spectrometer (AMS) (Canagaratna et al., 2007; Morgan et al., 2010; Allan et al., 2014), refractive BC from a Single Particle Soot Photometer (SP2) (Stephens et al., 2003; McMeeking et al, 2010; Allan et al., 2014) and aerosol size distribution from a Scanning Mobility Particle Sizer (SMPS) (Wang and Flagan, 1990; Morgan et al., 2015) and a GRIMM model 1.108 optical particle counter (OPC) (Heim et al., 2008). Further details about the instruments used aboard the Bae-146 during SAMBBA can be found in Sect. S2.1, including information about measurement uncertainty. See Allan et al. (2014) for more specific details regarding the aerosol sampling during SAMBBA.

The flights sampled a broad region spanning 46-68°W, 1-12°S (Fig. 2). Aerosol properties and fire emissions (Fig. 1) varied across this region, so we separated data into a western region (54-68.5°W, 6-12°S) and an eastern region (43-50°W, 4.5-15°S) following Johnson et al. (2016). We note that the aircraft sampling in the eastern region (including one full flight and sections of three flights) was limited relative to the sampling performed in the western region (including 14 full flights and sections of five flights). We used aircraft data from both vertical profiles and straight and level runs (SLR). To avoid bias, time periods when the plane was actively sampling smoke plumes were removed from the SLR data using a plume removal algorithm (Darbyshire et al., 2018). Visually observable plumes were specifically avoided when performing vertical profiles during SAMBBA so any enhancements due to smoke plumes in the profile data are small. Time periods when in-cloud sampling was performed were also filtered out of the data; specifically the data was screened for cloud artefacts when the liquid water content exceeded 0.05 g m$^{-3}$ (Darbyshire et al., 2018).

### 2.3.2 Ground observations

A large suite of instruments were deployed at a site in the southwest Amazon (8.69°S, 63.87°W) (Brito et al., 2014). The site is located in a forest reserve about 5 km from Porto Velho (population of around 500 000) and is usually upwind of the city (Brito et al., 2014). Here we used measurements from an Aerosol Chemical Speciation Monitor (ACSM; Ng et al. (2011))

and an Aethalometer (Magee Scientific, model AE30). The ACSM measured 30 min resolution mass concentrations of particulate ammonium, nitrate, sulfate, chloride, and organic species in the 75 - 650 nm size range. The Aethalometer measured 5 min resolution equivalent black carbon ($BC_{eq}$) mass concentrations. Details regarding the Aethelometer and ACSM measurement uncertainty can be found in Sect. S2.2. Data from both instruments are available from the 6th September to the 1st October 2012. Mean aerosol mass concentration (ACSM + Aethalometer) during this period was 13.7 µg m$^{-3}$, with OA contributing an average of 83% of total mass. Mean aerosol mass concentrations were lower in Phase 2 (6.0 µg m$^{-3}$) compared to Phase 1 (17.8 µg m$^{-3}$). Full details are provided in Brito et al. (2014). PM2.5 concentrations were measured using gravimetric filter analysis, with a measurement duration ranging from less than 1 day to ~7 days (Artaxo et al., 2013).

### 2.3.3 AERONET aerosol optical depth

We used spectral columnar AOD measured by Aerosol Robotic Network (AERONET) Cimel sun photometers (Holben et al., 1998) from 5 stations deployed across the region that have data available for the SAMBBA campaign period: Porto Velho UNIR (63.94°W, 8.84°S), Alta Floresta (56.10°W, 9.87°S), Rio Branco (67.87°W, 9.96°S), Cuiaba-Miranda (56.02°W, 15.73°S), Santa Cruz UTEPSA (63.20°W, 17.77°S). We used Version 3 Level 2 cloud-screened and quality assured daytime average AOD (Giles et al., 2019), retrieved at 500 nm. Locations of the AERONET stations are shown in Fig. 2.

### 2.3.4 MODIS aerosol optical depth

We used daily AOD retrieved at 550 nm from the Moderate Resolution Imaging Spectroradiometer (MODIS) instrument on board the Terra and Aqua satellites for the SAMBBA campaign period to calculate regional average AODs. Specifically, we used the Collection 5.1 Level-3 MODIS Atmosphere Daily Global Product gridded to 1°×1° resolution (Terra: MOD08_D3; Aqua: MYD08_D3; https://modis-atmosphere.gsfc.nasa.gov/products/daily) (Hubanks et al., 2008) acquired through NASA's Level 1 and Atmosphere Archive and Distribution System (LAADS) (https://ladsweb.modaps.eosdis.nasa.gov/). Daytime equator crossing is at 1030LT for Terra and at 1330LT for Aqua.

### 2.3.5 Measurement uncertainty

Section S2 describes further details of the instrumentation used during SAMBBA; including information about measurement calibration and uncertainty. In summary, for conditions during SAMBBA the mass concentration measurement uncertainty has been estimated to be: ~20% for the aethelometer (Schmid et al., 2006); 10-35% for the ACSM (depending on the species, OA is 15%; Crenn et al., 2015); ~30% for the AMS (Bahreini et al., 2009; Middlebrook et al., 2012); and ~30% for the SP2 (Schwarz et al., 2008; Shiraiwa et al., 2008). For AOD retrievals, the 1σ uncertainty is estimated to be ±0.05+15% for MODIS (Levy et al., 2010) and ±0.01 AERONET (Giles et al., 2019).

### 2.3.6 Comparing model and observations

To compare the model to the aircraft and ground-based observations, we linearly interpolated the simulated hourly data along the flight path of the aircraft and to the horizontal location of the Porto Velho ground station. To compare with the aircraft AMS and ground-based ACSM measurements, the same detection ranges of the instruments (see Sects. 2.3.1 and 2.3.2) were applied to the simulated mass concentrations. Prior to analysis, simulated data corresponding to periods of missing or invalid measurement data were removed. To quantify the agreement between model and observations, we use the Pearson correlation coefficient (r) and normalised mean bias factor (NMBF) as defined by Yu et al. (2006):

$$NMBF = \frac{(\sum M_i - \sum O_i)}{|\sum M_i - \sum O_i|}\left[\exp\left(\left|\ln\frac{\sum M_i}{\sum O_i}\right|\right) - 1\right]$$

where $M$ and $O$ represent model and observed values, respectively, for each time step, $i$. A positive NMBF indicates the model overestimates the observations by a factor of NMBF+1. A negative NMBF indicates the model underestimates the observations by a factor of 1−NMBF.

## 3. Results and Discussion

### 3.1 Surface aerosol measurements

Figure 3 shows surface PM2.5 concentrations observed at Porto Velho, in the southwest Amazon, from January to November 2012. Observed PM2.5 concentrations are less than 2 µg m$^{-3}$ between January to July 2012, increasing to 30-50 µg m$^{-3}$ in late August and September, then declining to less than in 10 µg m$^{-3}$ in October. This seasonal cycle is well reproduced by the model with all fire emission datasets. Simulated PM2.5 concentrations are enhanced by biomass burning from August through to October, when more than 80% of PM2.5 concentrations are from biomass burning. PM2.5 concentrations during September are well reproduced by the model with GFED and FINN emissions, but underestimated by the model with GFAS emissions. PM2.5 concentrations are underestimated during early August, potentially indicating that emission datasets have missed fires during the start of the dry season (see Fig. S2). During the SAMBBA campaign (13 Sep − 3 Oct), PM2.5 concentrations are well reproduced by the model with FINN (r$^2$=0.65; NMBF=0.03) and GFED (r$^2$=0.69; NMBF=-0.45) but underestimated with GFAS (r$^2$=0.44; NMBF=-1.09) (see Table 2 for a summary of NMBF values).

Figure 4 compares simulated and measured composition-resolved aerosol at Porto Velho during September 2012. Measured total aerosol mass, calculated as mass measured by the ACSM plus BC$_{eq}$ measured by the aethelometer, varies consistently with measured PM2.5 concentrations during the campaign (Fig. S6). However, when averaged over the gravimetric filter analysis sampling time, measured total (ACSM+BC$_{eq}$) aerosol mass concentrations are consistently lower than measured PM2.5 concentrations by ~20-60% (Fig. S6a). This difference in the measurements is mostly apportioned to the reduced aerosol detection-size range from the ACSM (i.e. submicrometric) in comparison to the gravimetric analysis (< 2.5 µm)

(Sect. 2.3.2), and, to a smaller extent, the different measurement techniques and aerosol species unaccounted by the on-line instrumentation (ACSM) e.g. crustal elements.

Average observed total (ACSM+$BC_{eq}$) aerosol mass concentrations are 20 µg m$^{-3}$ in Phase 1 reducing to 7 µg m$^{-3}$ in Phase 2. The model with fire emissions captures the decrease in observed aerosol mass concentrations between Phase 1 and Phase 2, but underestimates the magnitude of the reduction (5-10 µg m$^{-3}$, depending on the fire emission dataset, compared to 13 µg m$^{-3}$ in the observations). The model with GFED simulates observed total mass well in the ACSM detection range in Phase 1 (NMBF=0.08), but overestimates observed mass in Phase 2 (NMBF=0.89). Conversely, the model with GFAS emissions simulates observed total mass reasonably well in Phase 2 (NMBF=0.30) but underestimates in Phase 1 (NMBF= -0.49). The model with FINN emissions overestimates observed total mass in both Phase 1 (NMBF=0.45) and Phase 2 (NMBF=1.94).

Observed total aerosol mass is dominated by OA (84%), with BC contributing 9% and summed $NH_4$, $NO_3$ and Chl contributing less than 5% of total mass during the SAMBBA campaign (Fig. 4b). Simulated aerosol (with fire emissions included) is also dominated by OA (86-88%) with BC contributing a slightly smaller fraction of the total aerosol mass (5%) than observed (see Fig. S7 for simulated and measured hourly OA and BC time series). $NH_4$, $NO_3$ and Chl are not accounted for in GLOMAP. Sulfate accounts for 2.6% of the observed total aerosol mass during the campaign, but 5-11% in the model (Fig. 4b). Sulfate concentrations are well reproduced by the model with no fire emissions and are overestimated when fire emissions are included (Table 2). This suggests that either emissions of sulfate from fires are overestimated or that other sources of sulfate are overestimated in the model in the region of Porto Velho.

**3.2 Aerosol mass concentration vertical profile**

Figure 5 compares average vertical profiles of OA, sulfate and BC measured on the aircraft to that simulated by GLOMAP. As before the data is split into Phase 1 (flights 1–8) and Phase 2 (flights 9–20). We also split the data spatially into western and eastern Amazon regions (see Fig. 2). We note that the aircraft sampling in the eastern region was limited relative to sampling in the western region (Sect. 2.3.1). Figure S8 shows the number of OA (from the AMS) and BC (from the SP2) observations per vertical bin for the western region (Phases 1 and 2) and eastern region. Figure 5 shows that observed aerosol concentrations are greatest in the boundary layer (BL) then reduce rapidly above (see also Fig. S9). Figures 5 and S9 show that the shape of the aerosol vertical profile is well reproduced by the model, further confirming that simulated vertical mixing and the vertical injection height of fire emissions are reasonable. Observed aerosol concentrations are relatively constant between the surface and ~2500 m in the western Amazon and between the surface and ~4000 m in the eastern Amazon. This behaviour is reproduced by the model, and is likely due to a deeper BL over grassland vegetation in the eastern Amazon (Fig. S10).

In the western Amazon, average concentrations of OA below 2.5 km (roughly the BL) were 19 µg m$^{-3}$ in Phase 1 compared to 6 µg m$^{-3}$ in Phase 2, similar in both magnitude and temporal pattern to the surface observations at Porto Velho. In Phase 1, the model underestimates observed OA concentrations in the BL with all emission datasets (NMBF=-0.25 for FINN to -1.64

for GFAS). OA concentrations in Phase 1 in the western Amazon between 3 km and 4.5 km are also underestimated by the model with all emission datasets consistent with comparisons in the BL. The model does not simulate the observed reduction in OA concentrations between Phase 1 and Phase 2 overestimating OA concentrations in the western Amazon in Phase 2 with GFED (NMBF=0.39) and FINN (NMBF=1.21) emissions, but good agreement with GFAS (NMBF=0.02). This may be

because the emission datasets report only moderately lower emissions in Phase 2 compared to Phase 1 (Figure S 3a; Table 1), but also because the model may underestimate wet removal of aerosol during Phase 2 (consistent with model and observation comparisons in Archer-Nicholls et al. (2015)). In the eastern Amazon, average concentrations of OA below 4 km (roughly the BL) of 16 μg m$^{-3}$ are underestimated by the model with all three emission datasets (NMBF=-0.92 for GFAS to -3.14 for FINN).

Disagreement between observed and simulated OA may be due to uncertainty in the OA:OC ratio. In this study we assume an OA:OC ratio of 1.4, at the lower end of the range (1.4 to 2.6) assumed by other models (Tsigaridis et al., 2014). Philip et al. (2014) combined satellite data and AMS measurements to estimate an OA:OC ratio of 1.3 to 2.1. Preliminary analysis of aircraft data during SAMBBA suggests an OA:OC ratio of 1.5 to 1.8 for fresh BB aerosol and 2.0 to 2.3 for aged aerosol (Johnson et al., 2016). Assuming an OA:OC ratio of 2.3 would enhance our simulated OA concentrations by 60% reducing

our underestimate of OA in both the western (Phase 1: NMBF=-0.64 to 0.29) and eastern (NMBF=-1.59 to -0.20) Amazon.

Observed refractive BC (rBC) concentrations in the BL are ~1.0 μg m$^{-3}$ in the western Amazon during Phase 1 dropping to ~0.5 μg m$^{-3}$ during Phase 2. BC concentrations observed at the surface (Sect. 3.1; Phase 1, 1.6 μg m$^{-3}$; Phase 2, 0.9 μg m$^{-3}$) are greater than those measured on the aircraft, although this may be partly due to the different measurement techniques used and different size detection ranges. In the eastern Amazon, observed rBC concentrations are higher (1.8 μg m$^{-3}$), qualitatively

reproduced by the model with GFAS and GFED emissions but not with FINN emissions. The model strongly underestimates BC concentrations in the eastern Amazon with all emission datasets, particularly with FINN (NMBF=-1.48 for GFAS to -6.10 for FINN). In the west, the agreement is more variable, with the model well simulating concentrations in Phase 1 with FINN emissions but overestimating in Phase 2, GFED underestimating in Phase 1 but well simulating concentrations in Phase 2 and the model with GFAS emissions underestimating in both phases.

In the western Amazon, comparison with sulfate aerosol is fairly consistent with OA and BC comparisons, with the model underestimating in Phase 1 and overestimating in Phase 2. In the eastern Amazon, the model overestimates sulfate concentrations even without a contribution from fires, suggesting that other natural and anthropogenic sulfate sources may be overestimated in the model.

Observed average BC:OA mass concentration ratios in the BL vary from 0.05 in the western Amazon (Phase 1, 1/19=0.05; 
Phase 2, 0.5/6=0.08) to 0.11 (1.8/16=0.11) in the eastern Amazon. These ratios reflect the much higher BC emission factors found for flaming Cerrado fires in the eastern Amazon relative to tropical forest fires in the western Amazon (Hodgson et al., 2018). Simulated ratios are in good agreement with observations in the western Amazon with all emission datasets (e.g.

Phase 1, FINN: 0.9/15=0.06; Phase 2, FINN: 0.8/13=0.06). In the eastern Amazon, BC:OC ratios are underestimated using FINN (0.26/3.8=0.07) emissions, with better agreement using GFED (0.63/7.0=0.09) and GFAS (0.74/8.2=0.09) emissions.

In the western Amazon, average aerosol concentrations at 4 km are ~27-50% of concentrations in the BL (<2.5 km) ($OA_{P1}$: 7/19=0.37, $OA_{P2}$: 3/6=0.50; $BC_{P1}$: 0.27/1=0.27, $BC_{P2}$: 0.26/0.5=0.52). Marenco et al. (2016) reported a mean aerosol layer of 2.0±0.4 km during the SAMBBA campaign, which is consistent with the model results presented here. A plume-rise model coupled to WRF-Chem overestimated OA concentrations at 6-8 km altitude observed over tropical forest regions during SAMBBA, suggesting the plume rise model overestimated fire injection height (Archer-Nicholls et al., 2015). Injecting fire emissions into the surface layer (Fig S11, "GFED_surflev") has a relatively small impact on the simulated aerosol vertical profile in the west (a mean change of -0.3% below 2.5 km), with a small change in the bias against observations (e.g. for OA in Phase 1; NMBF= -0.82 for GFED and -0.77 for GFED_surflev), demonstrating that vertical mixing rapidly redistributes aerosol in the model. In the eastern region, the impact is larger (a mean change of +5% below 4 km), with a slight improvement in the model bias (e.g. for OA; NMBF= -1.23 for GFED and -0.90 for GFED_surflev). Overall there appears to be limited evidence for the need for substantial injection of fire emissions above the BL for fires in this region.

Overall, the comparisons with aircraft observations show variable agreement between model and observations. The model with GFAS emissions consistently underestimates observed aerosol mass concentrations by up to a factor 3, but gives the best agreement (relative to GFED and FINN) with observations in the eastern Amazon. Agreement between the model and observations with GFED and FINN emissions is more variable with up to a factor 2-3 underestimation or overestimation, depending on the region and time period. In general, the model with FINN emissions performs well against observations in the western Amazon in Phase 1 (when observed aerosol mass concentrations are relatively high), but gives the largest underestimation of aerosol mass concentrations in the eastern Amazon (relative to GFED and GFAS).

### 3.3 Aerosol size distribution

Figure 6 compares simulated aerosol size distributions against those measured on the aircraft during straight and level runs. In the eastern Amazon, the model underestimates particle number below 300 nm diameter, consistent with aerosol mass comparisons (Sect. 3.2). In the western Amazon, the model with FINN and GFED emissions generally well matches the observed size distribution above 200 nm diameter ($N_{200}$), with a small underestimate during Phase 1 with GFED ($NMBF_{P1}$=-0.52; $NMBF_{P2}$=-0.08) and a small overestimate during Phase 2 with FINN ($NMBF_{P1}$=-0.13; $NMBF_{P2}$=0.39) consistent with the vertical profiles of aerosol mass (Fig. 5). The model with GFAS emissions underestimates throughout the size distribution (e.g. for $N_{200}$: $NMBF_{P1}$=-1.20; $NMBF_{P2}$=-0.45), consistent with earlier comparisons.

There is a persistent underestimation of aerosol number at particle sizes below about 100 nm. We assume all biomass burning emissions are emitted into the accumulation mode with geometric mean diameter of 150 nm (Sect. 2.1.1; Mann et al., 2010), which is substantially higher than observed in the Porto Velho ground station data (94 nm; Brito et al., 2014). The observations suggest biomass burning makes a considerable contribution to aerosol number from ~50 to 200 nm diameter

that is not included in the model. This is consistent with Vakkari et al. (2018), where assumed emission size distributions in models poorly represented the number of particles in the 30–100 nm (Aitken mode) size range for southern African savannah and grassland fires.

We performed two sensitivity tests where we varied the assumed emission size distribution for primary biomass burning aerosol in GLOMAP (Fig. S12). Reid and Hobbs (1998) measured count median diameters (CMD) of 130±10 nm ($\sigma$=1.68±0.02) and 100±10 nm ($\sigma$=1.77±0.02) for deforestation fires and 100±10 nm ($\sigma$=1.91±0.15) for Cerrado fires (Reid et al., 2005). Assuming a CMD of 100 nm increases the simulated particle number concentration below 100 nm diameter by factors of ~1.8 (with $\sigma$=1.7) and ~1.5 (with $\sigma$=1.8) over the SAMBBA regions. This results in a reduction in the negative bias in simulated number concentration above 50 nm ($N_{50}$) (GFED (150 nm, $\sigma$=1.59): $NMBF_{WestP1}$=-1.85; GFED (100 nm, $\sigma$=1.7): $NMBF_{WestP1}$=-0.51), but a slight increase in the negative bias in $N_{200}$ (GFED (100 nm, $\sigma$=1.7): $NMBF_{WestP1}$=-0.55). Therefore, reducing the assumed emission size distribution for primary BC and OC from biomass burning may be important for cloud condensation nuclei concentrations, but will have a small effect on simulated aerosol mass and AOD (see Sect. 3.5.1).

### 3.4 Aerosol optical depth

Figure 7 compares simulated and satellite-retrieved (MODIS) AOD at 550 nm (AOD550) over the eastern and western regions for the SAMBBA campaign period. Compared to MODIS, the model generally underestimates AOD550 over both regions and with all fire emission datasets (NMBF=-2.41 to -0.38; Table 2). The model with GFAS emissions has the largest underestimation in the western Amazon (the smallest with FINN) and the model with FINN emissions has the largest underestimate in the eastern Amazon (the smallest with GFAS). The model with FINN emissions underestimates AOD550 in the western Amazon in Phase 2 even when it overestimates aerosol mass concentrations throughout the vertical profile. In the eastern region, the underestimation of AOD550 with all emission datasets is consistent with the comparison of the vertical profile of aerosol mass concentration.

Figure 8 compares simulated and observed AOD at 500 nm (AOD500) at five AERONET sites across western and southern Amazonia during SAMBBA (no data is available from the AERONET station located in the eastern region during the campaign). There is reasonable agreement between AOD500 reported by AERONET and AOD550 reported by MODIS, with AERONET generally reporting higher values (Aqua, NMBF=-0.60 to -0.27; Terra, NMBF=-0.47 to -0.18), partly due to differences in the wavelengths of the retrievals.

Consistent with comparisons to MODIS, the model generally underestimates AOD500 at all stations and with all fire emissions, except at two stations in the western Amazon (Rio Branco (in both campaign phases) and Porto Velho (in Phase 2)) with FINN emissions. The negative model bias in AOD500 across all AERONET stations is consistent with the negative model bias in AOD550 (against MODIS) (Table 2), but is smaller at some individual stations (Fig. 8). This is likely due to multiple reasons including differences in: i) the AOD wavelengths (500 nm versus 550 nm); ii) the AERONET and MODIS

retrieval uncertainties (Sect. S2.3); iii) the location/region of comparison, affecting magnitude and sources of AOD; and iv) the AERONET and MODIS data coverages.

For all stations, the model with GFAS emissions has the largest underestimation (Table 2). The fire emission dataset that gives the smallest model bias varies between GFED and FINN depending on the station and time period (Phase 1 or 2 of the campaign; Table 2). At Porto Velho, the model with GFED emissions underestimates AOD500 (NMBF=-0.70; Table 2) during Phase 2 even when it overestimates aerosol mass concentrations at the surface (NMBF=0.89; Table 2). The smallest model bias for all simulations is at Rio Branco in the western Amazon (e.g. with FINN: $NMBF_{P1}$ = 0.03, $NMBF_{P2}$ = 0.11).

Figure 9 compares average vertical profiles of aerosol scattering and extinction coefficients (at 550 nm) measured on the aircraft to that simulated by GLOMAP. In the western region, during Phase 1, the model underestimates the observed scattering coefficient throughout the vertical profile (NMBF=-1.29 with GFAS to -0.29 with FINN; Table 2); consistent with the comparisons against MODIS and AERONET AOD. During Phase 2, the agreement between simulated and measured scattering coefficient is more consistent with the vertical profile of aerosol mass concentrations than with MODIS or AERONET AOD; with good agreement or overestimation (NMBF=-0.07 with GFAS to 0.92 with FINN). The agreement between simulated and measured extinction coefficient is similar, but with larger negative biases during Phase 1 and slightly smaller positive biases in Phase 2. In the eastern region, the model strongly underestimates observed scattering (NMBF= -3.3 to -1.29) and extinction (NMBF= -3.90 to -1.51) coefficients, consistent with MODIS AOD550 (with larger negative biases than for total aerosol mass concentrations).

In summary, the model generally underestimates observed AOD during the SAMBBA campaign. The negative biases in simulated AOD and scattering and extinction coefficients are generally larger than in simulated aerosol size distribution (> 200 nm diameter) and in total aerosol mass concentrations at the surface (consistent with Reddington et al. (2016)) and aloft, which suggests that model underestimation of AOD is not solely due to an underestimation of biomass burning aerosol mass and/or emissions. The calculation of AOD also depends on aerosol optics and water uptake. We explore the sensitivity of simulated AOD to these other factors in the following section.

## 3.5 Exploring the sensitivity of simulated aerosol optical depth

In Reddington et al. (2016) we identified a greater model underestimation of AOD than surface PM2.5 in the Amazon region, where coincident observations were available, suggesting that the negative model bias in AOD could be caused by errors in the calculation of AOD rather than by errors in simulated aerosol properties. However, due to a lack of available observations, we were unable to rule out errors in simulated aerosol size distribution and vertical profile (i.e. an underestimation of aerosol aloft) with any certainty.

In this work, using the detailed SAMBBA observations, we have shown that the model well represents the vertical profile of aerosol mass concentrations and aerosol size distribution (in the diameter range relevant for visible light) in the western Amazon, yet continues to underestimate AOD. Below we explore the sensitivity of simulated AOD to the treatment of

biomass burning emissions in GLOMAP and to other relevant aerosol properties including aerosol mixing state, refractive indices and hygroscopicity (summarised in Table 3, Fig. 10 and Fig. S13). To quantify the sensitivity of each simulation we calculate the percentage change of simulated hourly AOD550 during the SAMBA campaign relative to simulation 1 in Table 3; the relative changes are summarised in Fig. 10. Figure S13 summarises the agreement between simulated and measured optical properties during the SAMBBA campaign for each simulation in Table 3.

### 3.5.1 Biomass burning aerosol emission strength, particle size and injection height

Figure 10 shows that simulated AOD is sensitive to the fire emission dataset used in the model. Changing between GFED emissions (simulation 1 in Table 3) and GFAS emissions (simulation 2 in Table 3) changes simulated hourly AOD550 during the SAMBBA campaign by -14% on average in the western region and by +10% on average in the eastern region (Fig. 10). Changing from GFED emissions to FINN emissions (simulation 3 in Table 3) increases simulated AOD550 in the western region (a mean change of +32%) and decreases simulated AOD550 in the eastern region (a mean change of -28%) (Fig. 10). As discussed in Sect 3.4, the model with FINN emissions has the smallest underestimate against MODIS AOD550 in the western region (the largest underestimation is with GFAS), but overestimates measured scattering and absorption coefficients during Phase 2 (Fig. S13). In the eastern region, the model with GFAS has the smallest underestimate of MODIS AOD550 and aircraft-measured scattering and absorption coefficients (the largest underestimation is with FINN) (Fig. S13).

Altering the injection height of biomass burning emissions to the surface level (simulation 4 in Table 3) has an almost negligible effect on simulated AOD550, with mean changes of <1% in the eastern region (ranging from +6 to -17% on the hourly timescale) and -3% in the western region relative to the control simulation (Fig. 10). Injecting GFED emissions at the surface results in an increase in the model NMBF in AOD550 against MODIS in the west (e.g. Phase 1: from 0.97 to -1.03) and a slight reduction in the NMBF in the east (from -1.44 to -1.42) (Fig. S13).

Reducing the assumed emission size for primary BC and OC particles from biomass burning (from a CMD of 150 nm to 100 nm; simulation 5 in Table 3) decreases simulated AOD550 in both the eastern (-9%) and western (-13%) regions (Fig. 10), consistent with the decrease in simulated $N_{200}$ (Sect. 3.3). As a result, the model bias in AOD550 against MODIS is increased relative to the control from NMBF=-0.97 to -1.28 in the west (Phase 1) and from NMBF= -1.44 to -1.70 in the east (Fig. S13).

### 3.5.2 Mixing state

We find that simulated AOD is relatively insensitive to the assumption about the aerosol mixing state; with less than 5% difference in the magnitude of AOD550 between internally mixed (simulation 1; Table 3) and externally mixed (simulation 6; Table 3) cases (consistent with Reddington et al., 2016). Calculating AOD550 assuming optical properties derived from an external mixture of aerosol species leads to slightly reduced values (by ~1-4%; mean reduction in the west: ~2%; mean reduction in the east: ~1%; Fig. 10) when compared to AOD550 calculated assuming an internal (volumetrically-averaged)

aerosol mixture (Fig. 10). Therefore, assuming an internal mixture leads to slightly improved agreement between simulated and observed AOD over the external mixture assumption (Fig. S13).

Han et al. (2013) also find relatively small changes in the magnitude of AOD (0.03 to 0.07) in high AOD regions (~0.8 to 2.0) between internally and externally mixed cases, with the internal mixture assumption giving higher values than the external mixture assumption. Curci et al. (2015) find a greater difference (~37%) in simulated AOD between internally and externally mixed assumptions, with the external mixed case giving the highest AOD. However, the greater sensitivity of simulated AOD to the mixing state assumption in Curci et al. (2015) was primarily due to the difference in the calculation of the aerosol number size distribution rather than the difference in the calculated optical properties. The GLOMAP model simulates both mass and number concentration of each size mode so the total number concentration stays identical for both mixing state assumptions (in the externally mixed case the number concentration of particles in a given size mode is split between aerosol components based on the volume fraction of that component in the mode). We note that the internally mixed case used in this study does not consider different mixing structure assumptions i.e. core-shell internal mixing, which may account for an additional uncertainty of ~5-10% in simulated AOD (Curci et al., 2015).

### 3.5.3 Refractive index

To investigate the sensitivity of simulated AOD to assumptions about the aerosol optical properties, we calculated AOD550 from the model simulation with GFED emissions assuming a range of refractive indices appropriate for BC and POM aerosol (see simulations 7 to 12 in Table 3). We find that the magnitude of simulated AOD550 varies by up to ~7% (relative to the control AOD550) depending on the choice of refractive indices.

Applying smoke aerosol refractive indices from Matichuk et al. (2007; 2008) to the model BC and POM components (simulations 8-10 in Table 3) leads to a small mean decrease in AOD550 relative to the control (by 2-5% in the eastern region and 0-3% in the western region; Fig. 10). Assuming medium and highly absorbing refractive indices for BC from Bond and Bergstrom (2006) (simulation 11 in Table 3) increases AOD550 by an average of 4-6% in the eastern region and 2-4% in the western region. Using the highly absorbing refractive index for BC (simulation 12 "rfidx_6" in Table 3) gives the best agreement between model and satellite-retrieved (MODIS) AOD550 out of the refractive index sensitivity tests (Fig. S13).

The relatively small sensitivity of simulated AOD to assumed aerosol refractive indices is consistent with previous studies (Matichuk et al., 2007; Curci et al., 2015; Reddington et al., 2016) and suggests that the negative bias in AOD cannot be wholly explained by the uncertainty associated with this assumption.

### 3.5.4 Aerosol water uptake

Aerosol water uptake plays a significant role in determining AOD, altering the refractive index and the size distribution of the aerosol. Our estimate of aerosol water uptake depends on the calculation method (including assumptions made regarding

aerosol hygroscopicity; described in Sects. 2.1.2 and S1), the model relative humidity (from ECMWF reanalyses) and the simulated aerosol physical/chemical properties (size distribution and composition).

To test the sensitivity of AOD to the calculation of aerosol water uptake, we compare AOD550 calculated using two methods (described in Sect. S1): 1. using ZSR online in the model (simulation 1 in Table 3); and 2. using the κ-Köhler water
uptake scheme (Petters and Kreidenweis, 2007) offline during post-processing (simulation 13 in Table 3). Our previous work demonstrated that simulated AOD is sensitive to this calculation, with simulated AOD440 varying by a factor of ~1.6 between the upper and lower estimates of water uptake (Reddington et al., 2016). We find the same here, with AOD550 varying by a mean factor of 1.6 over the western Amazon and a mean factor of 1.5 over the eastern Amazon, between the two calculation methods. Using the κ-Köhler water uptake scheme decreases AOD550 (by 32-39%; Fig. 10) relative to
AOD550 calculated using ZSR, thus increasing the negative model bias against observations (Fig. S13).

To explore the sensitivity to assumed κ values, we varied κ values separately for the sulfate and POM components in the model. Assuming a higher κ for sulfate (1.19 as for sulphuric acid; Petters and Kreidenweis, 2007) (simulation 14 "κK_2" in Table 3) results in simulated AOD550 being 26% and 33% lower on average than ZSR in the eastern and western regions, respectively (Fig. 10). Assuming a higher κ for both sulfate (1.19) and for POM (0.2) (simulation 15 "κK_3" in Table 3)
results in simulated AOD550 being a 23-25% lower than ZSR on average (Fig. 10). Figure S14 shows the change in simulated AOD550 due to assuming different κ values for sulfate and POM relative to the simulation using the κ-Köhler water uptake scheme with κ of 0.53 for sulfate and 0.1 for POM (simulation 1 "κK_1" in Table 3). Using high κ values for both sulfate and for POM (simulation 15 "κK_3" in Table 3) increases simulated AOD550 on average by 12-23%, improving agreement with MODIS AOD550, relative to simulation "κK_1" (Fig. S13). However, these high κ values are
likely to be unrealistically high for biomass burning aerosol (particularly for sulfate) and despite this, the model bias remains negative.

The higher AOD550 values calculated using the ZSR scheme can be explained by the steeper hygroscopic growth curve for biomass burning aerosol (at ambient relative humidity (RH) < 90%) when calculated with the GLOMAP ZSR scheme compared with the κ-Köhler scheme (see Fig. 13 of Johnson et al. (2016)). At ambient RH above 90% the opposite is true,
with a steeper growth curve for the κ-Köhler scheme (due to the RH-restriction of 90% applied in GLOMAP for ZSR; see Sect. S1.1). However, the model RH stays below 90% during the SAMBBA campaign (see Sect. 3.5.5). We note that AOD simulated with ZSR (assuming sulfuric acid and high water uptake for organics) is likely to be an upper estimate for water uptake. Our results confirm the large uncertainty present in the simulated AOD due to aerosol hygroscopicity.

### 3.5.5 Relative humidity

In Reddington et al. (2016) we discussed the potential sensitivity of simulated AOD to errors in ambient RH. In this work we are able to evaluate the model relative humidity against SAMBBA aircraft observations. The model captures the shape of the mean profile of observed relative humidity in both western and eastern regions (NMBF=-0.03); with small

over/underestimates in the BL in the western/eastern regions, respectively (western BL: NMBF=0.10; eastern BL: NMBF=-0.05) (Fig. S15). However, the model underestimates the variability in observed RH above ~1.5 km altitude in the west and below ~4 km altitude in the east. In particular, in the western region, the model does not capture the elevated observed RHs ($\geq 90\%$), which disproportionately affect aerosol water uptake and hygroscopic growth. This underestimation is likely due to the relatively coarse vertical and horizontal resolution of the model. A recent study by Haslett et al. (2019) found that high AODs observed by AERONET in southern West Africa could only be recreated by accounting for very elevated and variable RHs in the BL. They found that humid layers had a significant impact on AOD (particularly for RH > 98 %), resulting in a wet AOD more than 1.8 times the dry AOD (Haslett et al., 2019). This suggests that inadequate representation of sub-grid variability of RH may contribute to the model discrepancy in AOD.

To explore the sensitivity of model AOD to variability in RH, we forced the model RH to match either the mean or maximum observed RH in each vertical model level (on days with available aircraft data) and calculated the resulting water uptake using the κ-Köhler scheme (with $\kappa_{POM}$=0.1; $\kappa_{SO4}$=0.53; simulations 18-20 in Table 3). These sensitivity tests have mixed results on simulated AOD (see Figs. 10 and S14). Setting the model RH to the mean observed RH in each vertical level (simulation 18 "κK_RH" in Table 3) results in a small mean increase in simulated AOD550 in the east (by ~11%) and small mean decrease in the west (by ~8%) relative to AOD550 calculated using κ-Köhler and GLOMAP RH (simulation 13 "κK_1" in Table 3; see Fig. S13). Setting the model RH to the maximum observed RH in each vertical level (with a restriction of RH $\leq$ 99%; simulation 20 "κK_RHmax99" in Table 3) increases model AOD550 (by ~58-87% on average; Fig. S14) improving agreement with MODIS AOD550 (Fig. S13), but leads to overestimation of the observed aerosol scattering and extinction coefficients between 4 and 6 km altitude. Using the maximum observed RH but with a restriction of RH $\leq$ 96% (simulation 19 "κK_RHmax96" in Table 3) increases by ~20-58% on average, improving agreement with MODIS AOD550 relative to simulation "κK_1" (Fig. S13) and maintains good agreement with the relative vertical profiles of the scattering and extinction coefficients. However, the negative bias in AOD550 remains larger than for total aerosol mass concentrations (Table 2).

Although these sensitivity tests do not resolve the model discrepancy in model AOD, they demonstrate firstly, the large sensitivity of simulated AOD to the magnitude and variability of RH and secondly, that changes in RH can be important for simulated AOD even at RH < 90% (for instance, in the eastern region). Improving model representation of RH variability, whilst using an upper estimate for water uptake (e.g., the online ZSR scheme in GLOMAP), would likely bring the model bias in AOD more in-line with that of total aerosol mass concentration but this would require thorough testing in a high resolution model.

**3.5.6 Model spatial resolution**

The relatively coarse spatial resolution of the simulated aerosol and relative humidity (Sect 3.5.5) fields may also contribute to the model underestimation of AOD, due to underestimation of sub-grid variability (e.g. Weigum et al., 2016). Increasing

model spatial resolution has been shown to increase simulated AOD by ~11-13% (Bian et al., 2009; Weigum et al., 2016), depending on the initial and altered grid resolutions. However, we note that comparisons between simulated and aircraft observed aerosol mass concentrations suggest that the model captures observed spatial variations in aerosol mass concentrations reasonably well for the SAMBBA period, at least over the western Amazon region.

Model spatial resolution will also affect the model–measurement sampling uncertainty, which can be up to 50% for hourly time-resolution data (e.g. Schutgens et al., 2016a; 2017; Reddington et al., 2017). In our analysis we have strived to reduce spatial and temporal sampling errors as much as possible by: 1) running the model and using analysed meteorology for the same time period as the observations; 2) temporally co-locating model and measurement data points, removing time periods with missing or invalid measurement points from the model data (as discussed in Schutgens et al., 2016b) (and temporal
averaging for bias calculations and comparisons with aircraft measurements); and 3) spatially co-locating model data to observational data points using interpolation (and spatial averaging for comparisons with aircraft and MODIS observations). For comparisons with aircraft measurements, we have also attempted to reduce measurement representativeness error by removing in-plume and in-cloud sampling from the data where possible. We estimate remaining model–measurement sampling uncertainty to be up to ~30%, corresponding to monthly average model-measurement comparisons (Schutgens et
al., 2016a). A higher resolution model would be required to accurately quantify the model–measurement sampling uncertainty for this specific analysis and to explore the degree of sensitivity of AOD to the spatial resolution of simulated aerosol and relative humidity fields in detail.

## 3.6 Summary of AOD sensitivity simulations and uncertainties

Simulated AOD varies by more than a factor 2 across the different sensitivity simulations (Figs. 10 and S14). The different
emission inventories change simulated AOD by up to approximately ±30%. Altering assumed refractive indices changes simulated AOD by less than 8% and assumptions about external mixing change simulated AOD by less than 2%. Changes to assumptions controlling aerosol water uptake change simulated AOD by up to -40% (for assumptions regarding the water uptake calculation scheme) and up to +20% (for assumptions regarding the hygroscopicity parameter, κ). Uncertainty in the variability of RH changed simulated AOD by up to +87%. This analysis suggests that the largest uncertainties in simulated
AOD are associated with uncertainty in aerosol water uptake and model representation of relative humidity.

For the magnitude of AOD observed during the SAMBBA campaign, the uncertainty in the retrievals of AOD are approximately ±30% for MODIS AOD550 and <10% for AERONET AOD500 (see Sects. 2.3.5 and S2). Although the uncertainties in AOD retrievals are important to consider, they are smaller than the uncertainties associated with simulated biomass burning aerosol properties and AOD.

## 4. Conclusions

We have used surface, aircraft and satellite observations made during the SAMBBA field campaign in the southern Amazon during September and October 2012 to improve our understanding of biomass burning emissions. We apply three different biomass burning emission datasets (FINN, GFAS, GFED) in the GLOMAP global aerosol model. In fire-impacted regions of the Amazon, total annual aerosol emissions from fires (averaged over 2002-2012) vary by up to a factor 2.4 across these datasets, highlighting the large uncertainty in aerosol emissions from fires. In 2012, annual aerosol fire emissions were 4-47% less than the 2002-2012 mean in the western Amazon, but 30-6% greater than the long-term mean in the eastern Amazon. This reflects declining deforestation rate and associated fires in the western Amazon over this period (Reddington et al., 2015) and opposing trends in fires in the eastern Amazon (Andela et al, 2018).

During 2012, observed surface PM2.5 concentrations in the southern Amazon increased from ~2 µg m$^{-3}$ between January and July to 30-50 µg m$^{-3}$ in September then declined to less than 10 µg m$^{-3}$ in October. Observed aerosol mass (in the 75 - 650 nm diameter size range) in September was dominated by OA which accounted for 84% of total mass with BC accounting for 9% of mass. The model reproduced the observed seasonal cycle of aerosol concentrations, with ~54-78% of simulated PM2.5 concentrations originating from fire emissions during September 2012. Fires are the dominant source of PM2.5 across the region during the dry season.

In the western Amazon, where deforestation fires are the dominant fire type, agreement between simulated and observed aerosol mass concentrations in the BL is variable, depending on the fire emission dataset used both for OA (NMBF = -1.6 to +1.2) and BC (NMBF=-1.5 to +0.6). In this region we do not find evidence that aerosol emissions are systematically underestimated across all emission datasets. In the eastern Amazon, where grassland/savannah fires are dominant, GLOMAP underestimates OA (NMBF = -0.9 to -3.1) and BC (NMBF = -1.5 to -6.1) concentrations with all three emission datasets. This suggests that all emission datasets may underestimate aerosol emissions from grassland/savannah fires in the eastern Amazon, although we acknowledge the limited measurement sampling in this region relative to the western Amazon. We assume fire emissions have an OA:OC ratio of 1.4. Increasing our OA:OC ratio to 2.3, towards the upper end of that used in models (Tsigaridis et al., 2014) and matching aged aerosol observed in SAMBBA (Johnson et al., 2016), would improve the model-observation OA comparison in the eastern Amazon but would lead to an overestimate of OA in the western Amazon in some periods.

Comparisons of the simulated particle number size distribution against aircraft observations revealed a persistent underestimation of number concentrations of particles smaller than ~100 nm diameter. Reducing the assumed emission size of primary carbonaceous aerosol in the model improved agreement with observed number concentrations of particles less than 100 nm diameter, but increased the negative bias in simulated AOD. Assuming a bimodal emission size distribution for primary biomass burning aerosol may solve the model discrepancy in particle number concentrations below ~100 nm diameter, while retaining the simulated accumulation mode of biomass burning aerosol. Model underestimation of particle

number concentration of particles less than 100 nm will have implications for simulation of cloud condensation nuclei concentrations and simulated aerosol-cloud interactions.

Observed vertical profiles of aerosol mass concentrations were characterised by enhanced concentrations from the surface up to around 2 km altitude in the western Amazon and 4 km altitude in the eastern Amazon. In our model, we assume that all emissions from vegetation fires in this region are injected below 3 km altitude, with ~80% of emissions injected below 1 km. The model simulated a realistic relative vertical profile of aerosol mass concentrations, suggesting that our assumptions about injection height are valid. Our results further confirm that Amazon fires rarely inject emissions above the BL, as found by previous studies (Archer-Nicholls et al., 2015, Marenco et al., 2016).

The model generally underestimates AOD across the Amazon both in comparison to AERONET (NMBF=-1.3 to -0.4) and MODIS (NMBF =-2.4 to -0.4). In the eastern Amazon, the underestimation of aerosol mass concentrations through the vertical profile contributes to this underestimation of AOD. In the western Amazon, the model underestimates AOD even when the vertical profile of aerosol mass concentration is either well predicted or overestimated. This suggests that underestimation of AOD may be due to uncertainties in the calculation of AOD, rather than underestimation of aerosol mass concentrations. To explore this possibility we tested the impact of uncertainty in refractive index, aerosol mixing state, aerosol water uptake and relative humidity on model AOD. We found that simulated AOD was most sensitive to assumptions about water uptake and the model representation of variability in relative humidity, leading to an average uncertainty range in simulated AOD of approximately -40% to +90%.

Overall, our work suggests that aerosol emissions from fires are on average underestimated over the Amazon, particularly over grassland/savannah fires in the eastern Amazon, albeit by less than the factor ~3-5 assumed in some previous studies. Confirming our previous work (Reddington et al., 2016) we find that simulated and observed aerosol mass concentrations are generally in better agreement than simulated and observed AOD. We show that the model underestimates AOD even when it reproduces the observed vertical profile of aerosol mass. This suggests that uncertainties in the calculation of AOD, rather than the aerosol mass concentration, are the dominant reason for underestimation of AOD, and we find largest sensitivity to uncertainty in water uptake and model representation of relative humidity variability. We therefore caution against using comparison with AOD to scale particulate emissions from fires, as has been done in a number of previous studies.

### Data availability

Data from all GLOMAP model simulations and processing codes are available from the corresponding author on request. All raw time series data from the FAAM research aircraft are publically available from the Centre for Environmental Data Analysis website, where the entire SAMBBA dataset may be accessed. Data masks for categorising flight patterns into plume-sampling and other sampling types (vertical profiles and SLRs) are currently available on request from Hugh Coe.

Biomass burning emissions datasets are available publically and can be accessed at the following web pages: http://bai.acom.ucar.edu/Data/fire/ (FINN); https://apps.ecmwf.int/datasets/data/cams-gfas/ (GFAS); https://www.globalfiredata.org/data.html (GFED). AERONET and MODIS aerosol optical depth data are available publically from NASA: https://aeronet.gsfc.nasa.gov/new_web/aerosols.html (AERONET) and https://ladsweb.modaps.eosdis.nasa.gov (MODIS).

**Author contributions**

C.L.R. performed the model simulations and analysed the model data. W.T.M., E.D., J.B., H.C. and P.A. provided aircraft and ground-based measurement data for model evaluation and provided expertise on aerosol measurements in the Amazon region. W.T.M., E.D. and J.B. were involved in the collection, processing and analysis of these datasets; PA and HC coordinated and planned the SAMBBA field campaign (with other co-principle investigators). D.V.S., J.M. and C.E.S. provided scientific input to the manuscript and all authors contributed to scientific discussions. C.L.R. and D.V.S. wrote the manuscript with input from all authors.

*Acknowledgements.* This research was supported by funding from the Natural Environment Research Council for the South American Biomass Burning Analysis (SAMBBA) project (number NE/J009822/1). The authors gratefully acknowledge the principal investigators (B. Holben and P. Artaxo) and their staff responsible for establishing and maintaining the five AERONET stations used in this study and providing quality-assured data.

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

**Tables**

| | Annual emission (2002-2012 mean) (Gg a⁻¹) | | | 2012 / (2002-2012) | SAMBBA emission (Gg day⁻¹) | |
|---|---|---|---|---|---|---|
| | OC | BC | OC:BC | OC+BC | OC | BC |
| Western Amazon (54-68.5°W, 6-12°S) | | | | | | |
| GFAS | 429 | 56.4 | 7.6 | 0.53 | 2.32; P1: 3.32; P2: 1.42 | 0.305; P1: 0.436; P2: 0.187 |
| FINN | 1060 [2.5] | 118 [2.1] | 9.1 | 0.77 | 8.69 [3.7]; P1: 11.9; P2: 5.81 | 0.958 [3.1]; P1: 1.31; P2: 0.637 |
| GFED | 546 [1.3] | 63.3 [1.1] | 8.6 | 0.96 | 5.02 [2.2]; P1: 8.21; P2: 2.11 | 0.580 [1.9]; P1: 0.944; P2: 0.249 |
| Eastern Amazon (43-50°W, 4.5-15°S) | | | | | | |
| GFAS | 336 | 46.6 | 7.2 | 1.46 | 5.64 | 0.796 |
| FINN | 181 [0.5] | 20.4 [0.4] | 8.8 | 1.30 | 2.39 [0.4] | 0.267 [0.3] |
| GFED | 223 [0.7] | 29.9 [0.6] | 7.5 | 1.94 | 5.23[0.9] | 0.693 [0.9] |

**Table 1.** Comparison of organic carbon (OC) and black carbon (BC) emissions from biomass burning in the GFASv1.2, FINNv1.5 and GFED4.1s emissions inventories. Emissions are summed separately for the western (54-68.5°W, 6-12°S) and eastern (43-50°W, 4.5-15°S) Amazon as shown in Fig. 2. Table reports long-term (2002-2012) mean annual total emissions; the ratio of annual total emissions in 2012 to the 2002-2012 mean; and total emissions for the SAMBBA campaign period (13 September – 3 October 2012). All values are given to 3 significant figures. The ratios (FINN:GFAS and GFED:GFAS) of total annual emissions during the 2002-2012 period and mean daily emissions during the SAMBBA campaign are given in parentheses. For the western Amazon, daily emissions are also shown for the two phases of the campaign: P1 (13 – 22 September) and P2 (23 September – 3 October).

|  | noBBA | FINN | GFED | GFAS |
|---|---|---|---|---|
| **Western Amazon, Phase 1** | | | | |
| PVH PM2.5 | -4.80 | **-0.09** | -0.54 | -1.35 |
| PVH Total mass | -3.43 | 0.45 | **0.08** | -0.49 |
| PVH OA | -3.45 | 0.50 | **0.11** | -0.50 |
| PVH BC | -53.12 | **0.02** | -0.34 | -1.32 |
| PVH Sulfate | **0.47** | 2.69 | 2.32 | 1.96 |
| Aircraft Total mass (<2.5 km) | -6.83 | **-0.23** | -0.76 | -1.42 |
| Aircraft OA (<2.5 km) | -7.63 | **-0.25** | -0.82 | -1.64 |
| Aircraft BC (<2.5 km) | -37.23 | **-0.13** | -0.78 | -1.52 |
| Aircraft Sulfate (<2.5 km) | -2.31 | **-0.48** | -0.72 | -0.83 |
| Aircraft Scattering (total column) | -4.62 | **-0.29** | -0.77 | -1.29 |
| Aircraft Extinction (total column) | -5.47 | **-0.37** | -0.89 | -1.46 |
| AOD550 (MODIS) | -5.25 | **-0.51** | -0.97 | -1.43 |
| AOD500 (AERONET) | -6.95 | **-0.47** | -0.53 | -1.26 |
| **Western Amazon, Phase 2** | | | | |
| PVH PM2.5 | -2.26 | 0.32 | **-0.25** | -0.61 |
| PVH Total mass | -0.72 | 1.94 | 0.89 | **0.30** |
| PVH OA | **-0.44** | 2.44 | 1.18 | 0.47 |
| PVH BC | -34.63 | **0.12** | -0.51 | -1.51 |
| PVH Sulfate | **-0.06** | 2.22 | 1.51 | 1.24 |
| Aircraft Total mass (<2.5 km) | -1.24 | 1.13 | 0.36 | **0.02** |
| Aircraft OA (<2.5 km) | -1.16 | 1.21 | 0.39 | **0.02** |
| Aircraft BC (<2.5 km) | -20.94 | 0.56 | **-0.06** | -0.56 |
| Aircraft Sulfate (<2.5 km) | -0.58 | 0.71 | 0.40 | **0.24** |
| Aircraft Scattering (total column) | -0.85 | 0.93 | 0.34 | **-0.07** |
| Aircraft Extinction (total column) | -1.07 | 0.83 | 0.26 | **-0.00** |
| AOD550 (MODIS) | -3.68 | **-0.38** | -0.70 | -1.06 |
| AOD500 (AERONET) | -4.50 | **-0.41** | -0.68 | -1.11 |
| **Eastern Amazon** | | | | |
| Aircraft Total mass (<4 km) | -8.43 | -2.60 | -1.00 | **-0.78** |
| Aircraft OA (<4 km) | -13.17 | -3.14 | -1.23 | **-0.92** |

| | | | | |
|---|---|---|---|---|
| Aircraft BC (<4 km) | -41.29 | -6.11 | -1.91 | **-1.48** |
| Aircraft Sulfate (<4 km) | **0.29** | 0.69 | 1.40 | 1.22 |
| Aircraft Scattering (total column) | -8.81 | -3.30 | -1.49 | **-1.29** |
| Aircraft Extinction (total column) | -10.80 | -3.90 | -1.75 | **-1.51** |
| AOD550 (MODIS) | -4.92 | -2.41 | -1.44 | **-1.23** |

**Table 2.** Summary of comparison between model and observations expressed as normalised mean bias factor (NMBF, blue indicates model underestimation). Comparisons are shown for observations from the Porto Velho measurement station (PVH): PM2.5 (particulate matter with diameters smaller than 2.5 µm) mass, total aerosol mass (mass measured by the ACSM plus equivalent black carbon measured by the aethelometer), and organic aerosol (OA), black carbon (BC) and sulfate mass concentrations; the aircraft: total aerosol mass (mass measured by AMS plus refractive BC measured by the SP2), and OA, BC and sulfate mass concentrations, aerosol scattering, and aerosol extinction; satellite: aerosol optical depth at 550 nm (AOD550) from MODIS; and AOD500 from AERONET. Aircraft comparisons are for concentrations below 2.5 km (western Amazon) or 4 km (eastern Amazon). AERONET comparisons are the average NMBF across 5 stations. For aircraft total aerosol mass, scattering and extinction, model and observations are compared only during time periods with available AMS measurements. Values are shown for the model with FINN1.5, GFAS1.2, GFED4.1s emissions and with no biomass burning emissions (noBBA). The numbers highlighted in bold show the model simulation with the smallest bias.

| # | Name | Description | Fire emissions | Refractive indices (RI) | | Mixing (internal/external) | Water uptake scheme | RH fields |
|---|---|---|---|---|---|---|---|---|
| | | | | BC | POM | | | |
| 1 | GFED | Control (GFED emissions) | GFED | $1.750 - 0.442286i$ ($\Lambda$=542 nm)[a] | $1.500 - 0.00i$ (all $\Lambda$)[a] | Internal | ZSR[b] | GLOMAP (ECMWF) |
| 2 | FINN | FINN emissions | FINN | $1.750 - 0.442286i$ ($\Lambda$=542 nm) [a] | $1.500 - 0.00i$ (all $\Lambda$) [a] | Internal | ZSR[b] | GLOMAP (ECMWF) |
| 3 | GFAS | GFAS emissions | GFAS | $1.750 - 0.442286i$ ($\Lambda$=542 nm) [a] | $1.500 - 0.00i$ (all $\Lambda$) [a] | Internal | ZSR[b] | GLOMAP (ECMWF) |
| 4 | surflev | GFED emissions injected into the model surface level | GFED | $1.750 - 0.442286i$ ($\Lambda$=542 nm) [a] | $1.500 - 0.00i$ (all $\Lambda$) [a] | Internal | ZSR[b] | GLOMAP (ECMWF) |
| 5 | emsize | GFED emissions with a reduced emission size for BC/OC particles (CMD=100 nm, $\sigma$=1.7) | GFED | $1.750 - 0.442286i$ ($\Lambda$=542 nm) [a] | $1.500 - 0.00i$ (all $\Lambda$) [a] | Internal | ZSR[b] | GLOMAP (ECMWF) |
| 6 | extmix | External mixing assumption. | GFED | $1.750 - 0.442286i$ ($\Lambda$=542 nm) [a] | $1.500 - 0.00i$ (all $\Lambda$) [a] | External | ZSR[b] | GLOMAP (ECMWF) |
| 7 | rfidx_1 | RI calculated for young smoke aerosol over southern Africa. | GFED | $1.54 - 0.025i$ ($\Lambda$=550 nm) [c] | $1.54 - 0.025i$ ($\Lambda$=550 nm) [c] | Internal | ZSR[b] | GLOMAP (ECMWF) |
| 8 | rfidx_2 | RI retrieved by Ndola AERONET station in Zambia; close to smoke sources (Sep 2000 mean) | GFED | $1.51 - 0.024i$ ($\Lambda$=440 nm) [d] | $1.51 - 0.024i$ ($\Lambda$=440 nm) [d] | Internal | ZSR[b] | GLOMAP (ECMWF) |
| 9 | rfidx_3 | RI retrieved by Ndola AERONET station in Zambia (16 Sep 2000) | GFED | $1.52 - 0.019i$ ($\Lambda$=440 nm) [d] | $1.52 - 0.019i$ ($\Lambda$=440 nm) [d] | Internal | ZSR[b] | GLOMAP (ECMWF) |
| 10 | rfidx_4 | RI retrieved by AERONET station, Jaru Reserve in Brazil (20 Sep 2002) | GFED | $1.50 - 0.02i$ ($\Lambda$=440 nm) [e] | $1.50 - 0.02i$ ($\Lambda$=440 nm) [e] | Internal | ZSR[b] | GLOMAP (ECMWF) |
| 11 | rfidx_5 | Mid-range value for RI for light absorbing carbon | GFED | $1.85 - 0.71i$ ($\Lambda$=550 nm)[f] | [set to $1.500 - 0.000i$][a] | Internal | ZSR[b] | GLOMAP (ECMWF) |
| 12 | rfidx_6 | Upper limit of RI for light absorbing carbon | GFED | $1.95 - 0.79i$ ($\Lambda$=550 nm)[f] | [set to $1.500 - 0.000i$][a] | Internal | ZSR[b] | GLOMAP (ECMWF) |
| 13 | $\kappa$K_1 | $\kappa$-Köhler to calculate aerosol water uptake. | GFED | $1.750 - 0.442286i$ ($\Lambda$=542 nm)[a] | $1.500 - 0.00i$ (all $\Lambda$)[a] | Internal | $\kappa$-Köhler ($\kappa_{OA}$=0.1[g]; $\kappa_{SO4}$=0.53[h]) | GLOMAP (ECMWF) |
| 14 | $\kappa$K_2 | $\kappa$-Köhler to calculate aerosol water uptake (using $\kappa$ for $H_2SO_4$) | GFED | $1.750 - 0.442286i$ ($\Lambda$=542 nm)[a] | $1.500 - 0.00i$ (all $\Lambda$)[a] | Internal | $\kappa$-Köhler ($\kappa_{OA}$=0.1[g]; $\kappa_{SO4}$=1.19[h]) | GLOMAP (ECMWF) |
| 15 | $\kappa$K_3 | $\kappa$-Köhler to calculate aerosol water uptake (using high $\kappa$ for OA and $\kappa$ $H_2SO_4$) | GFED | $1.750 - 0.442286i$ ($\Lambda$=542 nm)[a] | $1.500 - 0.00i$ (all $\Lambda$)[a] | Internal | $\kappa$-Köhler ($\kappa_{OA}$=0.2[i]; $\kappa_{SO4}$=1.19[h]) | GLOMAP (ECMWF) |
| 16 | $\kappa$K_4 | $\kappa$-Köhler to calculate aerosol water uptake (using CCN-derived $\kappa$ for $(NH_4)_2SO_4$) | GFED | $1.750 - 0.442286i$ ($\Lambda$=542 nm)[a] | $1.500 - 0.00i$ (all $\Lambda$)[a] | Internal | $\kappa$-Köhler ($\kappa_{OA}$=0.1[g]; $\kappa_{SO4}$=0.61[h]) | GLOMAP (ECMWF) |
| 17 | $\kappa$K_5 | $\kappa$-Köhler to calculate | GFED | $1.750 - 0.442286i$ | $1.500 - 0.00i$ | Internal | $\kappa$-Köhler | GLOMAP |

| | | | | $(\Lambda=542 \text{ nm})$[a] | (all $\Lambda$)[a] | | $(\kappa_{OA}=0.2$[i]; $\kappa_{SO4}=0.61$[h]) | (ECMWF) |
|---|---|---|---|---|---|---|---|---|
| | | aerosol water uptake (using high κ for OA and CCN-derived κ for $(NH_4)_2SO_4$) | | | | | | |
| 18 | κK_RH | κ-Köhler to calculate aerosol water uptake with mean aircraft RH vertical profile. | GFED | $1.750 - 0.442286i$ $(\Lambda=542 \text{ nm})$[a] | $1.500 - 0.00i$ (all $\Lambda$)[a] | Internal | κ-Köhler $(\kappa_{OA}=0.1$[g]; $\kappa_{SO4}=0.53$[h]) | Aircraft (mean RH) |
| 19 | κK_RH max96 | κ-Köhler to calculate aerosol water uptake with maximum aircraft RH vertical profile (capped at 96%). | GFED | $1.750 - 0.442286i$ $(\Lambda=542 \text{ nm})$[a] | $1.500 - 0.00i$ (all $\Lambda$)[a] | Internal | κ-Köhler $(\kappa_{OA}=0.1$[g]; $\kappa_{SO4}=0.53$[h]) | Aircraft (max RH, capped at 96%) |
| 20 | κK_RH max99 | κ-Köhler to calculate aerosol water uptake with maximum aircraft RH vertical profile (capped at 99%). | GFED | $1.750 - 0.442286i$ $(\Lambda=542 \text{ nm})$[a] | $1.500 - 0.00i$ (all $\Lambda$)[a] | Internal | κ-Köhler $(\kappa_{OA}=0.1$[g]; $\kappa_{SO4}=0.53$[h]) | Aircraft (max RH) |

[a] Bellouin et al., 2011; [b] Stokes and Robinson, 1966; [c] Haywood et al., 2003; [d] Matichuk et al., 2007; [e] Matichuk et al., 2008; [f] Bond and Bergstrom, 2006; [g] Gunthe et al., 2009; [h] Petters and Kreidenweis, 2007; [i] Petters et al., 2009.

**Table 3.** Summary of tests performed to explore the sensitivity of simulated aerosol optical depth (AOD) to assumptions about biomass burning emissions, aerosol optical properties (refractive indices (RI)), aerosol mixing state and aerosol water uptake calculation (Sect. 3.5). The water uptake schemes used (ZSR and κ-Köhler) are described in Sect. S1 ($\kappa$ is the component-specific hygroscopiticy parameter).

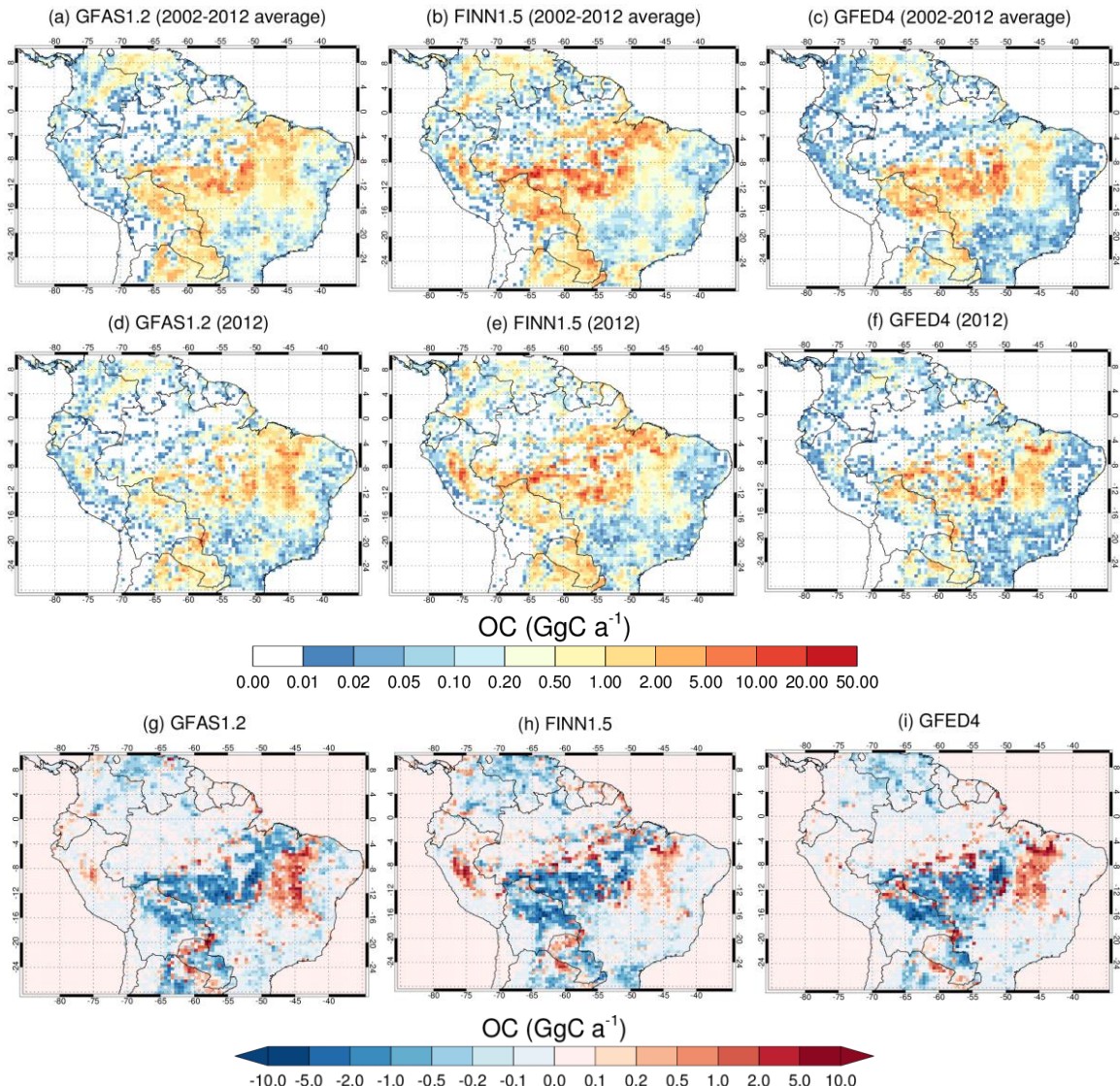

**Figure 1.** Maps of estimated total annual organic carbon (OC) aerosol emissions from fire shown as **(a)**-**(c)** an average for the period 2002 to 2012 and **(d)**-**(f)** for 2012. The difference in OC emissions between 2012 and 2002 to 2012 (2012 emissions minus 2002-2012 average) is shown in **(g)**-**(i)**. Blue colours show where emissions in 2012 were less than in 2002 to 2011, while red colours show where emissions were greater. Emissions are shown for: GFAS version 1.0 (2002-2011) and version 1.2 (2012) (left); FINN version 1.5 (middle); and GFED version 4 (right). The GFAS, FINN and GFED OC emissions were re-gridded onto a common grid of 0.5°x0.5° resolution for comparison.

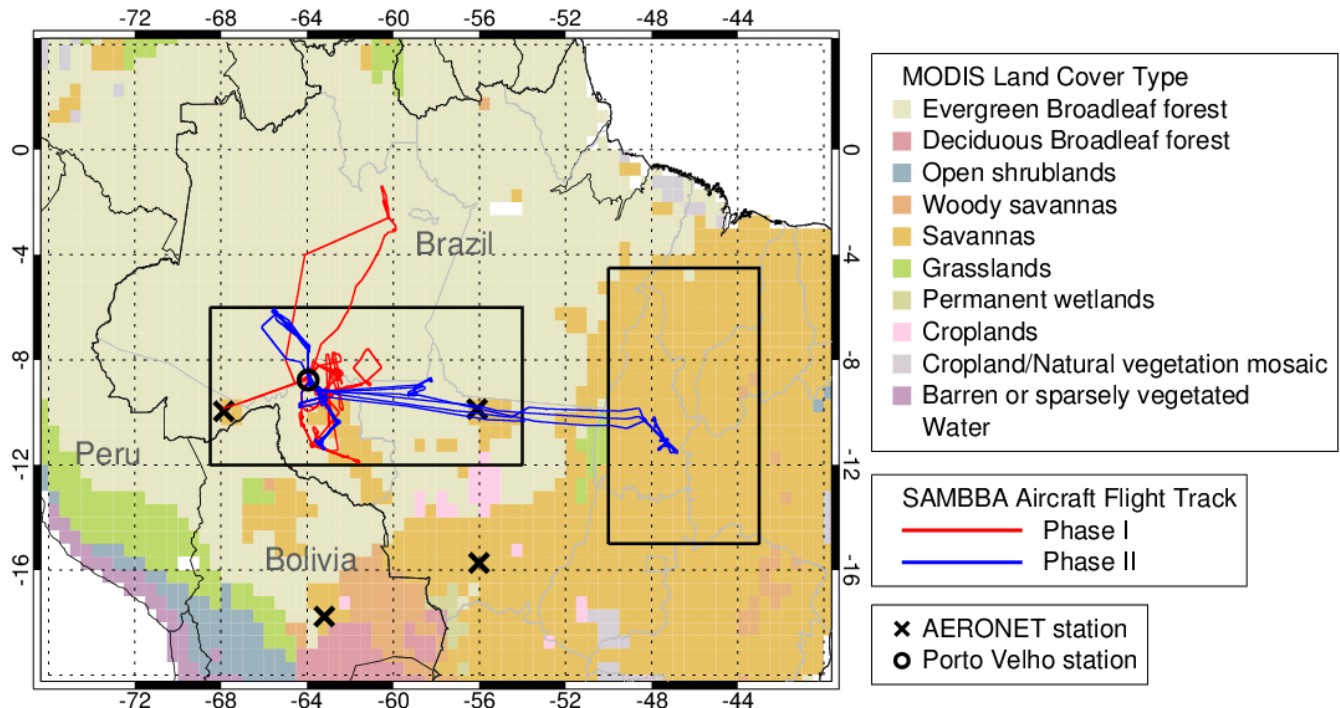

**Figure 2.** Flight tracks of the FAAM aircraft during the SAMBBA field campaign (Phase 1: 13 - 23 September; Phase 2: 23 September – 3 October 2012). The location of the Porto Velho ground station is shown by a black circle. The locations of AERONET stations operating during the SAMBBA campaign are shown by black crosses: Porto Velho UNIR (63.94°W, 8.84°S), Alta Floresta (56.10°W, 9.87°S), Rio Branco (67.87°W, 9.96°S), Cuiaba-Miranda (56.02°W, 15.73°S), Santa Cruz UTEPSA (63.20°W, 17.77°S). The eastern (43-50°W, 4.5-15°S) and western (54-68.5°W, 6-12°S) domains are shown with black boxes. Land cover type is shown using the standard MODIS land cover type data product (MCD12Q1) in the IGBP Land Cover Type Classification (Channan et al., 2014; Friedl et al., 2010).

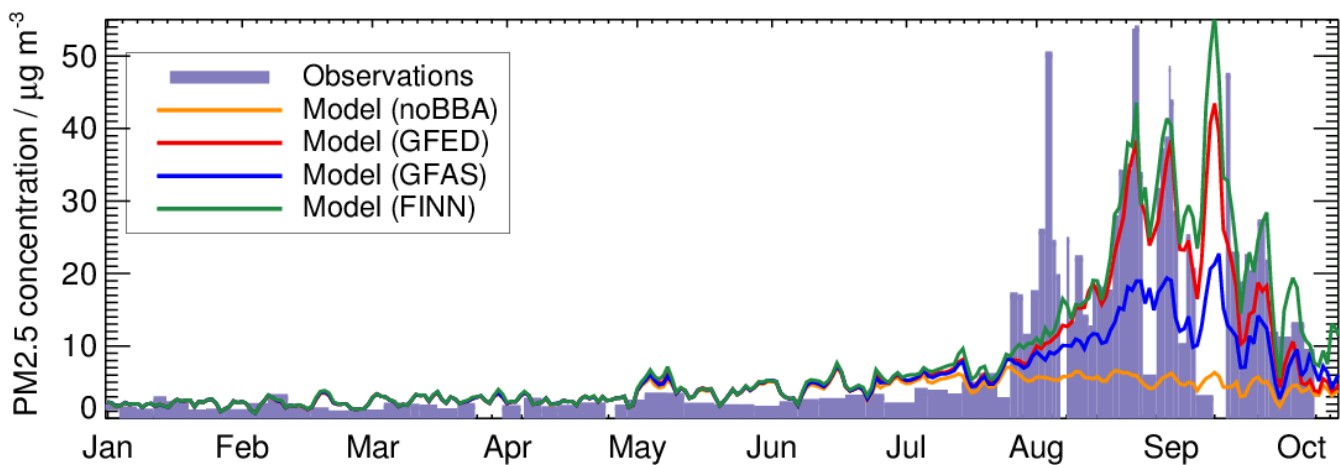

**Figure 3.** Time series of simulated (lines) and observed (bars) PM2.5 concentrations at Porto Velho between January and November 2012. Simulated daily mean concentrations are shown with FINN1.5 (green), GFAS1.2 (blue), GFED4 (red) emissions and with no biomass burning emissions (noBBA; orange). Observed PM2.5 concentrations are averages over sampling periods that ranged from <1 day to 7 days in 2012. The NMBF values are given separately for Phase 1 and Phase 2 of the SAMBBA field campaign in Table 2.

**(a)**

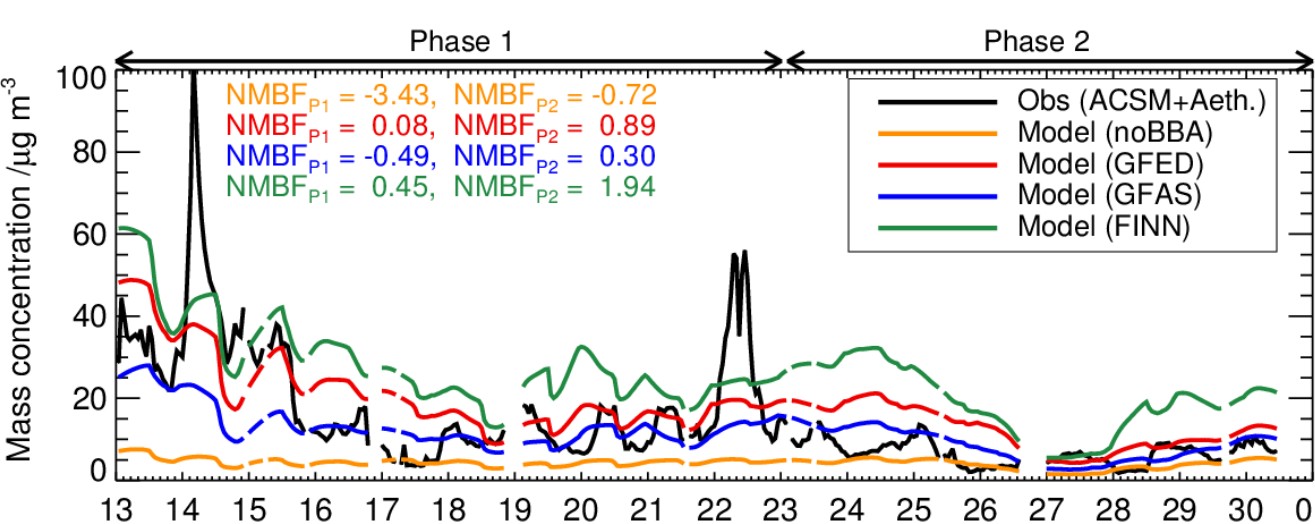

**(b)**

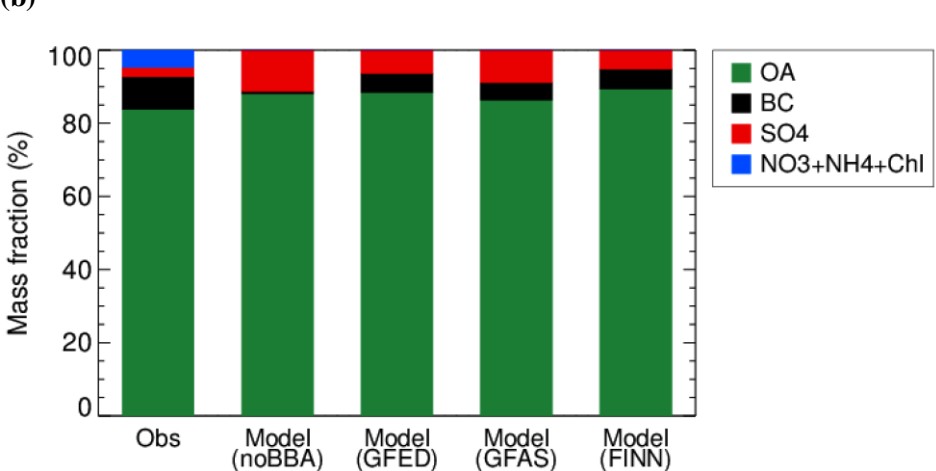

**Figure 4.** Composition resolved aerosol mass at Porto Velho ground station during the SAMBBA campaign. **(a)** Time series of hourly-mean observed (black) and simulated (colour) total aerosol mass. The observed aerosol mass is the total mass from the ACSM plus equivalent BC from the aethelometer. Simulated total aerosol mass is shown for the model with FINN1.5 (green), GFAS1.2 (blue), GFED4 (red) emissions and with no biomass burning emissions (noBBA; orange). Numbers on the panel show the NMBF for the SAMBBA campaign separately for Phase 1 (P1) and Phase 2 (P2) (also see Table 2). **(b)** Bar chart showing observed and simulated average aerosol composition during the campaign: black carbon (BC; black), nitrate+ammonium+chloride (NO3+NH4+Chl; blue, not treated by the model), organic aerosol (OA; green) and sulfate (SO4; red).

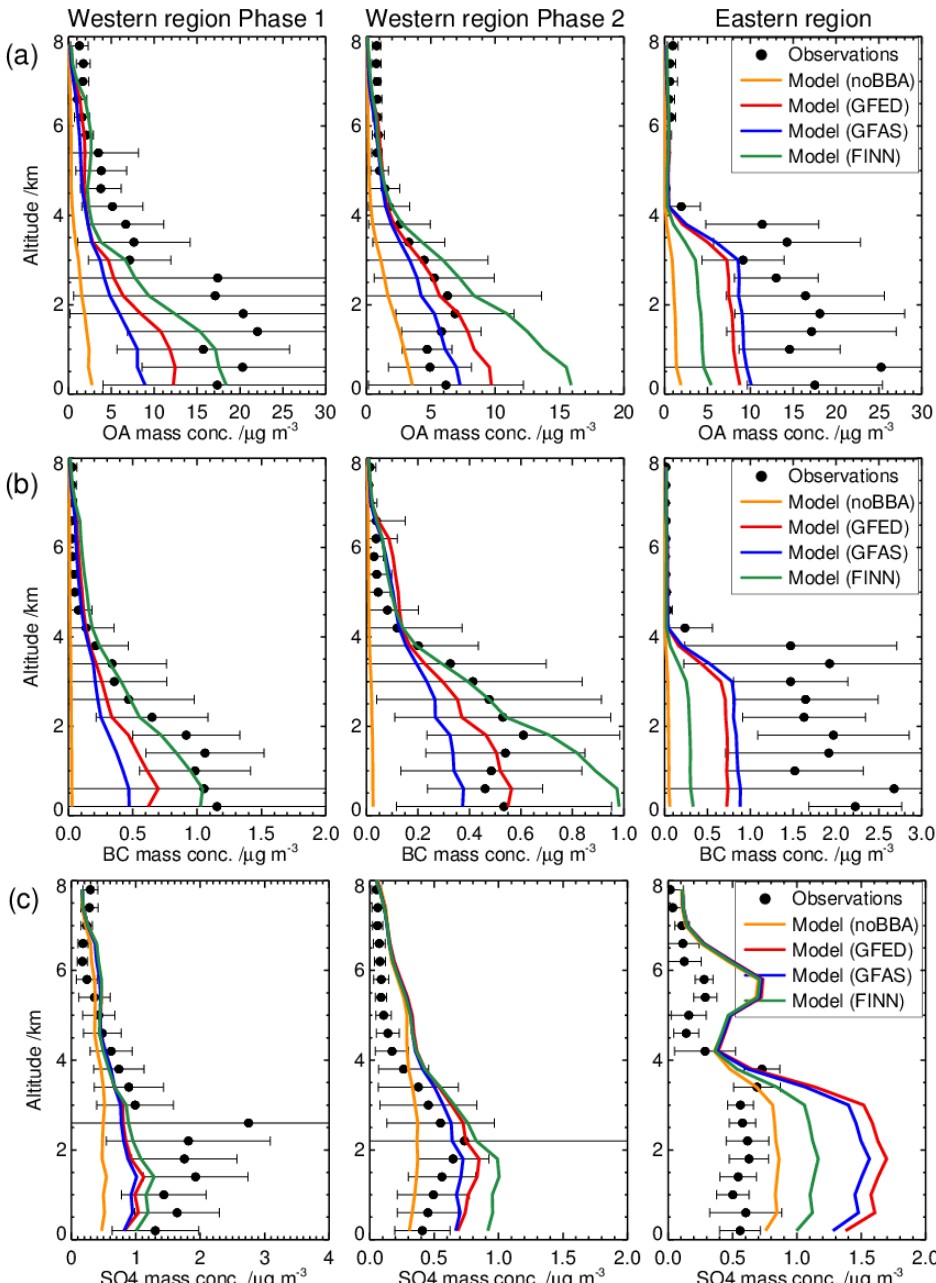

**Figure 5.** Mean observed and simulated vertical profiles of **(a)** organic aerosol (OA), **(b)** black carbon (BC) and **(c)** sulfate (SO4) during the SAMBBA aircraft campaign, sectioned into 400 m altitude bins. Observations are shown by the black data points; simulated concentrations are shown for the model with FINN1.5 (green), GFAS1.2 (blue), GFED4 (red) emissions and with no biomass burning emissions (noBBA; orange). The simulated data (linearly interpolated to the flight track of the

aircraft) and the observations are split into western and eastern regions of the Amazon (Fig. 2) and by time (Phase 1: 13/09/2012 – 22/09/2012, Phase 2: 23/09/2012 - 03/10/2012) for the western region. Error bars show the standard deviation of the observed mean. Concentrations are reported at standard temperature and pressure (STP) conditions (at 273.15 K and 1013.25 hPa). The NMBF values are given separately for the western region (Phase 1 and Phase 2) and eastern region in

Table 2.

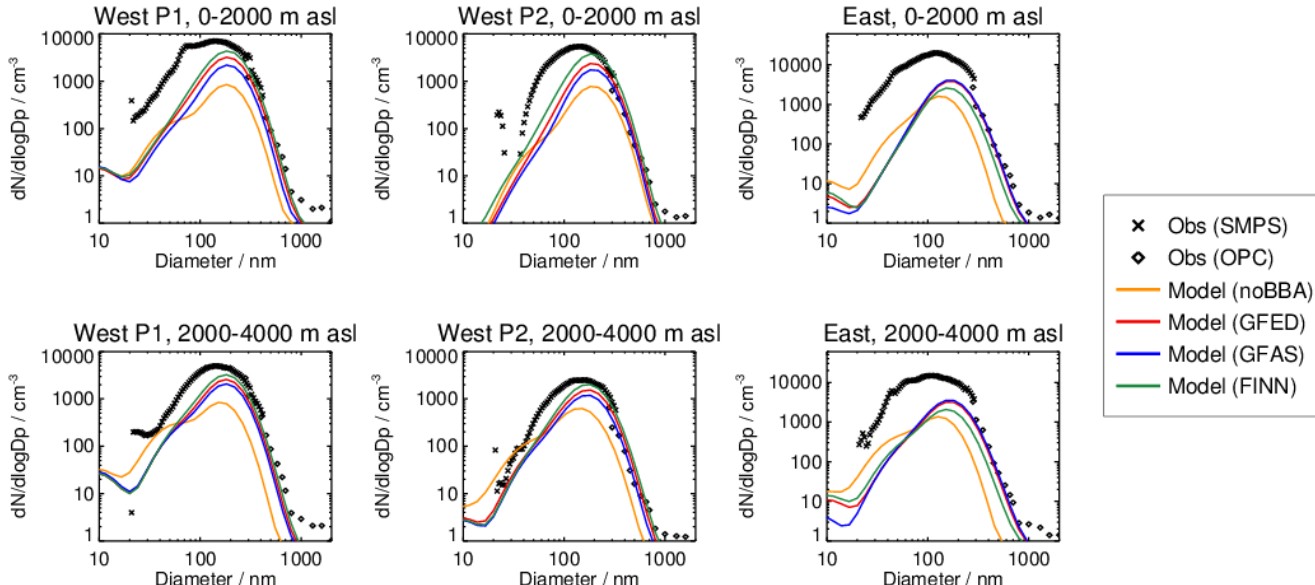

**Figure 6.** Mean observed (black) and simulated (colour) aerosol number size distributions during the SAMBBA aircraft campaign for two altitude bands: between the surface and 2 km (top panel) and between 2 and 4 km asl (bottom panel). The observed number size distribution was measured with Scanning Mobility Particle Sizer (SMPS; black crosses) and a Grimm optical particle counter (OPC; black diamonds). The simulated data (linearly interpolated to the flight track of the aircraft) and the observations are split into western and eastern regions of the Amazon (Fig. 2) and by time (P1: 13/09/2012 –

22/09/2012, P2: 23/09/2012 - 03/10/2012) for the western region. Simulated concentrations are shown for the model with FINN1.5 (green), GFAS1.2 (blue), GFED4 (red) emissions and with no biomass burning emissions (noBBA; orange).

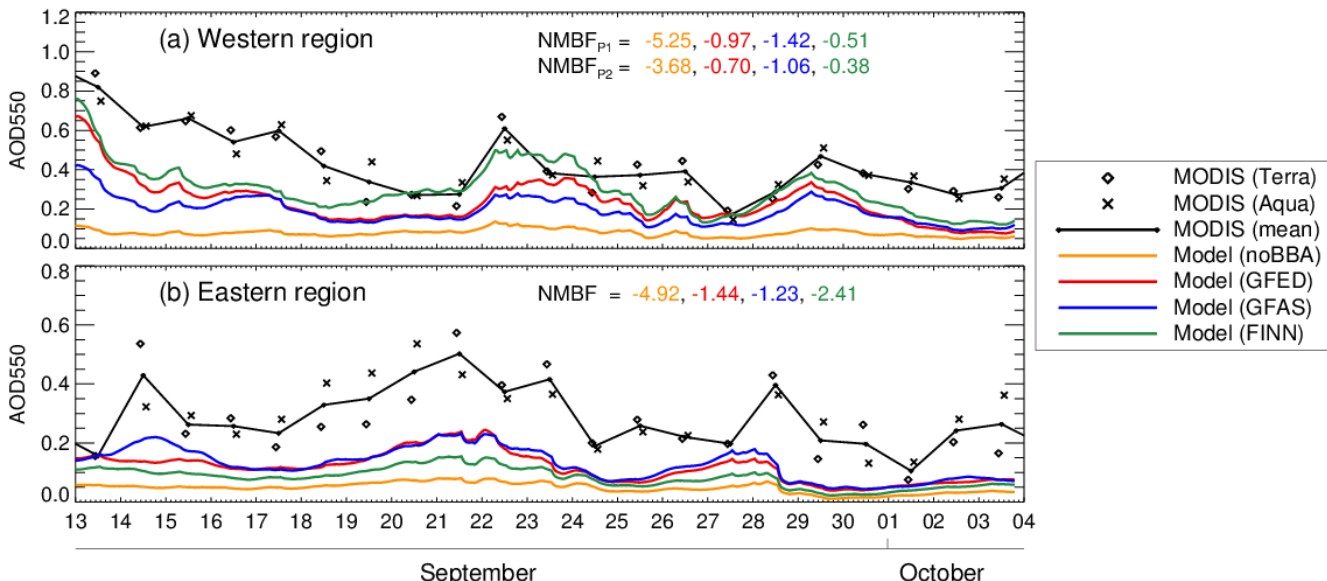

**Figure 7.** Time series of simulated (colour) and observed (black) aerosol optical depth at 550 nm (AOD550) for the SAMBBA campaign (13 September – 3 October 2012) over **(a)** the western Amazon (54-68.5°W, 6-12°S) and **(b)** the eastern Amazon (43-50°W, 4.5-15°S). Observed AOD550 retrieved by MODIS on-board Terra (over pass time: 10:30 local time) is shown by the black diamonds and AOD550 retrieved by MODIS on-board Aqua (over pass time: 13:30 local time) is shown by the black crosses; the black line shows an average value for each day (plotted at midday local time). Simulated hourly AOD550 (plotted at local time for Rondônia, Brazil: UTC-4h) is shown for the model with FINN1.5 (green), GFAS1.2 (blue), GFED4 (red) emissions and with no biomass burning emissions (noBBA; orange). The NMBF values are given separately for the western region (Phase 1 and Phase 2) and eastern region in Table 2.

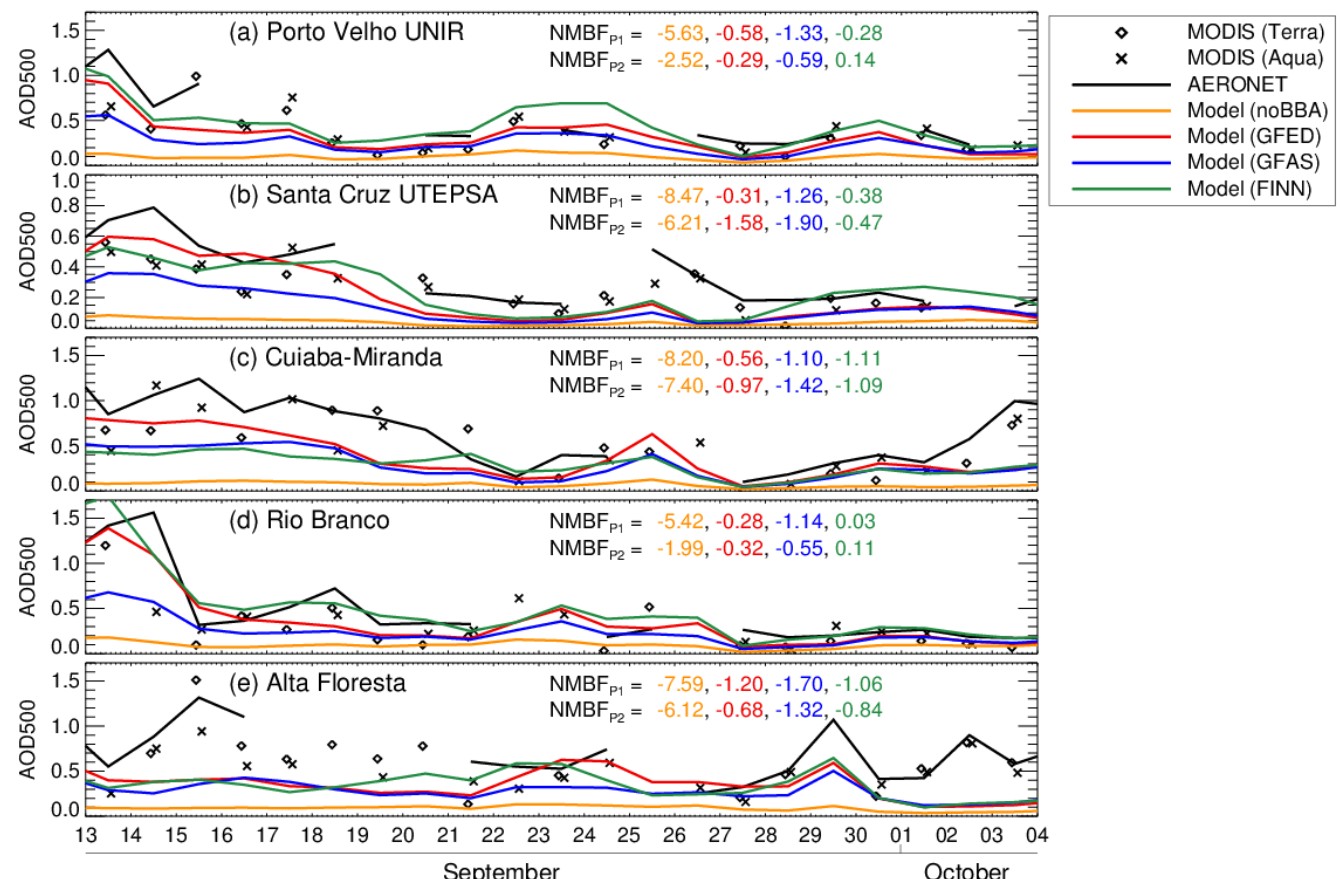

**Figure 8.** Time series of simulated (colour) and observed (black) daily aerosol optical depth (AOD) during the SAMBBA campaign at 5 AERONET stations in the western and southern Amazon: **(a)** Porto Velho UNIR (63.94°W, 8.84°S), **(b)** Santa Cruz UTEPSA (63.20°W, 17.77°S), **(c)** Cuiaba-Miranda (56.02°W, 15.73°S), **(d)** Rio Branco (67.87°W, 9.96°S) and **(e)** Alta Floresta (56.10°W, 9.87°S). Daily mean AOD at 500 nm (AOD500) from AERONET (black line) is compared to AOD550 retrieved by MODIS (Terra: black diamonds; Aqua: black crosses), using grid cells nearest the AERONET station location. Simulated daily mean AOD500 is shown for the model with FINN1.5 (green), GFAS1.2 (blue), GFED4 (red) emissions and with no biomass burning emissions (noBBA; orange). The NMBF values are given separately for Phase 1 (P1) and Phase 2 (P2) of the campaign.

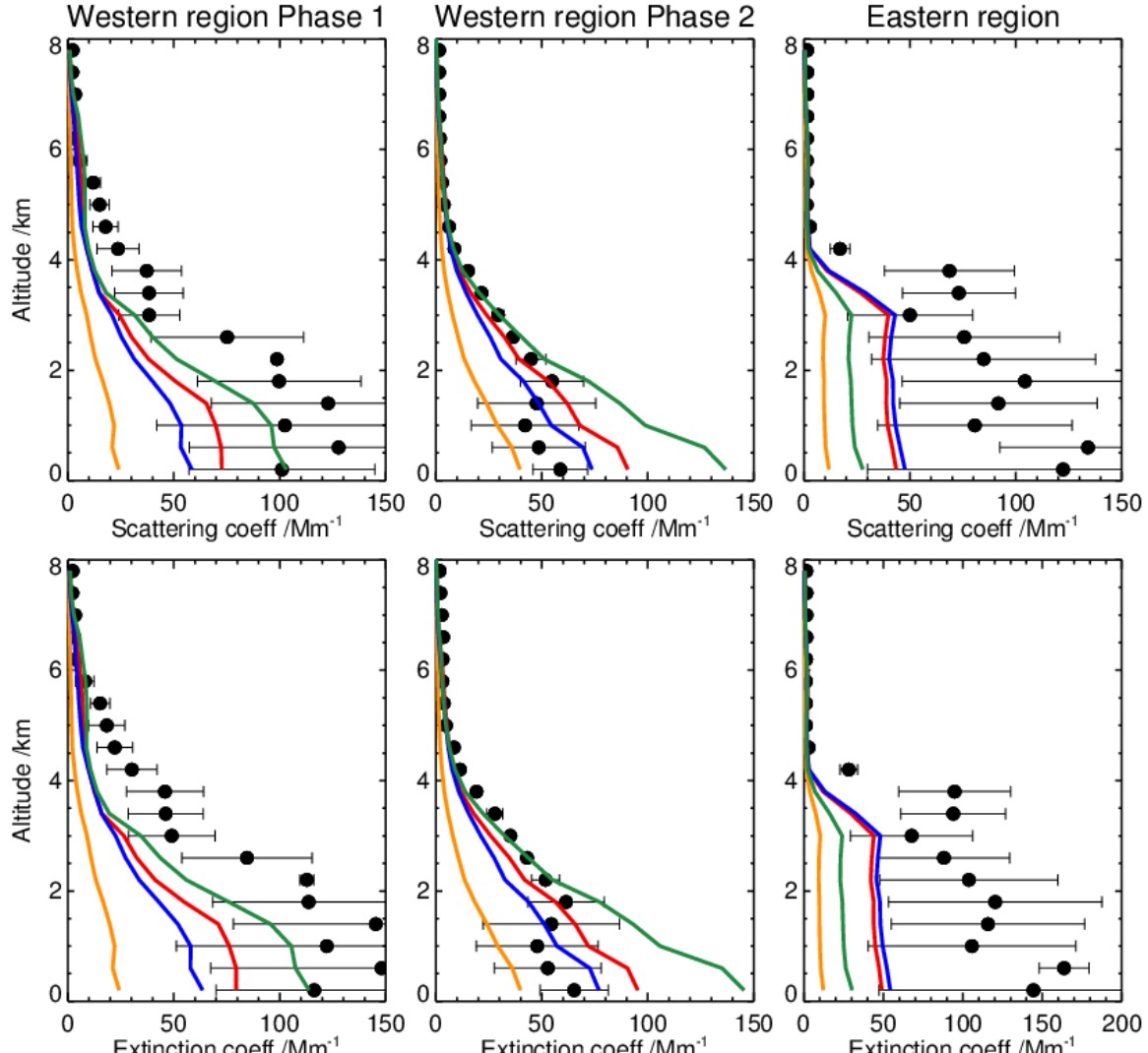

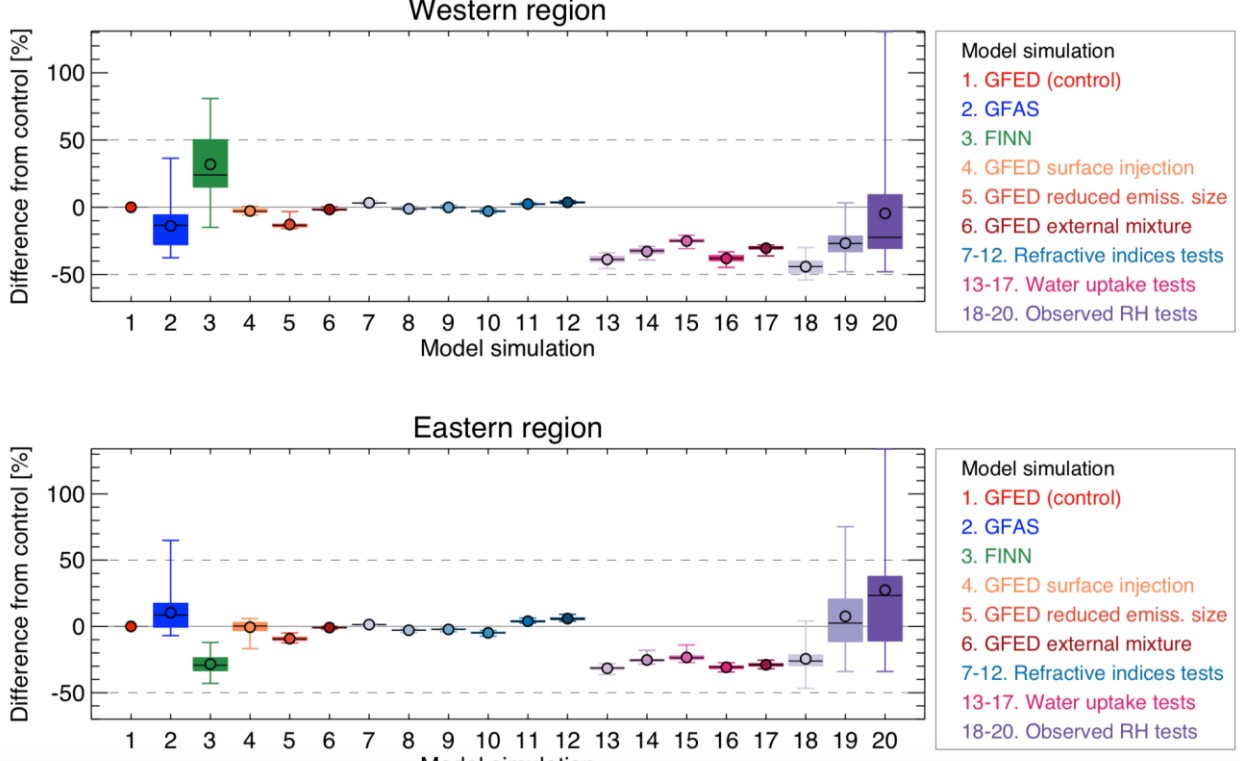

**Figure 10.** Box and whisker plot summarising the relative difference between simulated hourly mean AOD550 from the control simulation (GFED emissions with ZSR water uptake scheme) and each of the sensitivity simulations listed in Table 3. Simulation numbers in the figure correspond to the simulation numbers in Table 3. Simulations are compared during the SAMBBA campaign period (13 September – 3 October 2012) for the western region (54-68.5°W, 6-12°S; top panel) and the eastern region (43-50°W, 4.5-15°S; bottom panel) separately. Simulations 18, 19 and 20 are compared to the control simulation only on days with available aircraft measurements of RH. Circles show the mean values; whiskers show the minimum and maximum values; boxes show the 25th and 75th percentiles; and horizontal lines show the median values.