# Peer review of "Biomass burning aerosol over the Amazon: analysis of aircraft, surface and satellite observations using a global aerosol model"

_Atmospheric Chemistry and Physics, 2018_

## Referee Comment (RC1) · Anonymous Referee #1 · 8 Oct 2018

This study uses a combination of modeling (with various emission inventories) and observations to explore the importance of biomass burning aerosol over the Amazon. The study does a nice job of bringing together different observations and the comparisons against various inventories are enlightening (if only to understand how very uncertain biomass burning emissions are!). There are a few areas that should be improved to enhance the robustness of the study and its utility to the community before final publication:

1. The authors discuss a number of factors that could influence their comparison with observations, but disappointingly, they don't perform any sensitivity simulations to test

these. It would be nice to see the impact of (a) injecting all fire emissions at the surface and (b) decreasing the size of emitted biomass burning aerosol in the model on their results. This would enable the authors to be conclusive about these factors; without this the discussion remains largely unsubstantiated.

2. Observational uncertainties. The authors should include some information on the uncertainties associated with the various measurement techniques in Section 2.3 and include these uncertainties in their discussion of the measurement-model comparisons. In particular, aethalometer observations are highly uncertain with significant filter loading artifacts. The authors should acknowledge this and discuss what impact it might have on their results.

3. Are there sufficient number of observations in the "eastern region" to be statistically representative? It is hard to see from Figure 2 how many flights extend over this region and it might be useful to include the number of observations per vertical bin in Figure 5. The authors should acknowledge the limited sampling here.

4. Section 3.2: The authors show that the model fails to capture the decrease in aerosol concentrations from Phase 1 to Phase 2. Did the authors explore what role, if any, meteorology might play (temperature, wind direction, precipitation)? The emission inventory may not adequately reflect changes in burn conditions.

5. Figures 7 & 8, and page 11, lines 3-11: The authors suggest that Figures 7 and 8 are consistent, but they do not appear to be so. Figure 7 shows a clear underestimate in mean MODIS AOD over the western and eastern Amazon by all of the models, by a factor of ~2. Figure 8 shows that at least some of the models adequately capture (and sometimes overestimate!) the AOD observed at AERONET sites and by MODIS at these sites, with the exception of the early part of the Alta Floresta record. Therefore, the statement on page 11, line 7 "the model consistently underestimates..." is clearly false. Given the reasonable agreement between AERONET and MODIS at these sites, does this analysis suggest that the MODIS AOD observed in other regions of the Amazon is biased high? The authors need to correct their conclusions and discuss this more fully.

6. Section 3.5.3: The authors make a clear case that uncertainties associated with water uptake have a large impact on simulated AOD, but appear to show that this cannot explain the discrepancy between observations and their model (if they use an alternate approach the bias gets worse). Thus, as I read this, the authors fail to come up with any explanation for why simulated AOD appears biased low when mass concentrations are captured by their model. If so, the authors should be clear that this is not resolved and modify their conclusions and abstract accordingly. If they are saying that the uncertainty associated with water uptake could increase the AOD, they should show this result.

Other Minor Comments/Corrections

1. Page 2, Line 2: It seems that the goal of this study is to quantify the impacts of biomass burning emissions, not the emissions themselves (i.e. this isn't an inverse modeling study and the authors did not present a best-estimate of emissions from fires in the region). Please re-phrase.

2. Section 2.1: The model description is missing a few items: (a) the year of meteorology, (b) a description of aerosol removal (wet & dry), and (c) production totals of biogenic SOA for the region to compare to Table 1 emissions (also: did the simulation not include isoprene SOA?? This seems like an oversight).

3. Section 2.2: The authors focus their comparison of inventories on the OC emissions. They should comment somewhere in the text on whether the differences in BC emission (shown in Figure 1) are similar.

4. Page 5, line 29 & page 6, line 9: typo: Figure 2

5. Page 6, lines 12-13: briefly describe the main features of the plume removal algorithm, and how much data was removed using this filtering.

[Figure]

6. Section 2.3.4: include more information on the retrieval and relevant product including appropriate references.

7. Page 7, lines 28-30: This text is confusing. Figure 4a shows maximum concentrations of 100 ug/m3 (not 30-40 ug/m3) – did the authors mean to refer to Figure 3 here? Also the ACMS+BCeq in Figure 4 is not "consistently lower" than PM2.5 in Figure 3 as it is clearly higher on Sep 14 and 22. Please correct this text.

8. Page 8, lines 1-3: might differences in measurement technique (beyond size cut-offs) also be a factor?

9. Page 8, lines 13-15: how much of simulated OA is biogenic SOA?

10. Page 8, line 14: is the NH4+NO3+Chl contribution relatively consistent in the observations or are there days where these species make a larger contribution to total measured PM2.5?

11. Page 8, lines 13-15: how well does GLOMAP capture the speciated mass concentrations? i.e. what is the R2 between simulated and observed OA and BC?

12. Figure 5, 6, 7, 8: would be more legible if the authors included a legend

13. Page 8, line 25: Is the boundary layer deeper in the eastern Amazon than western Amazon in the model? Please comment on this in the text.

14. Page 9, line 12: typo "$\sim$ 0.5 ug/m3 during Phase 2"

15. Page 9, lines 24-29: Do the differences in observed BC:OA mass concentrations indicate anything about differences in fuel type or burn conditions in the western vs. eastern Amazon?

16. Page 10, line 11: why use only "straight and level runs"? How many measurements are included in these averages?

17. Page 13, lines 9-12: Have the authors compared the highest RH? These are the
values that will disproportionately impact water uptake and aerosol growth. How well does the model capture observed RH > 90%?

---

## Referee Comment (RC2) · Anonymous Referee #2 · 9 Oct 2018

**Review of Biomass burning aerosol over the Amazon: analysis of aircraft, surface and satellite observations using a global aerosol model**

**Carly L. Reddington et al.**

**Summary:**

Biomass burning is a major source of particulate matter pollution, regionally and globally. This has important implications for air quality and climate. Over South America, intense fires occur in August-September typically, providing the dominating source of atmospheric aerosols to the region. Yet considerable uncertainties in the magnitude of fire emissions remain. As such, the paper focuses on 'improving understanding of aerosol emissions from vegetation fires' by considering three different fire emission datasets (namely GFED4.1, GFAS1.2 and FINN1.5) to account for fire emissions in their modelling work. The authors used a global aerosol model (GLOMAP) to study how the simulated particulate matter (PM) concentration and aerosol optical thickness (AOT) are affected by the three different fire emission inventories. These results are compared against a comprehensive set of surface, aircraft and satellite observations collected over the Amazon region during September 2012. The authors have highlighted the spatial and temporal variation in the three different fire emissions and how it affects simulated quantities. Overall, the authors conclude that GLOMAP has skill in predicting reasonable surface concentration and vertical profile of PM over South America despite noticeable differences between the emission inventories. However, GLOMAP simulated AOT is found to be systematically underestimated. The authors therefore recommend caution when evaluating global models using AOTs to constrain particulate emissions from fires.

**General comments:**

The work presented here shares a lot of its DNA with a previous publication from the same group (i.e. Reddington et al., 2016). In this previous effort, the authors used the same modelling framework to argue that GLOMAP showed better agreement with observed PM mass concentration compared to AOT, potentially suggesting that some of the discrepancy between top-down and bottom-up studies may be connected to the calculation of AOT. In the present work, the authors test this hypothesis further by: *i*) providing a much more detailed evaluation of the GLOMAP model simulations against a comprehensive set of observations collected over South America during the SAMBBA campaign, and *ii*) performing a model sensitivity analysis exploring the assumptions related to the calculation of AOT in GLOMAP.

The paper is well structured and reads easily. The model evaluation is rigorous and convincing. The figures are clear and illustrate the points made in the manuscript. This paper is interesting and has a good potential. The last section of the result (Sect. 3.5) however feels too rushed in its current form and could benefit from the support of more visual material (i.e. show some plots for these results). The results in this section are mentioned too briefly, and do not provide a critical interpretation that would ensure more generally applicable results that could be subsequently transferred to other atmospheric aerosol models. The paper is quite weighted towards model evaluation. A model evaluation is only really useful if it used to interpret observed relationships or processes. As such, section 3.5 fails to convincingly demonstrate the assumption tested in this study.

The diversity in fire emissions highlighted here has important implication for aerosol modelling over this region, and likely in any region influenced by biomass burning. Contrasting the uncertainties from emissions with the uncertainties related to AOT calculation could really improve the scientific strength of the work. I would suggest reworking section 3.5, and perhaps add a discussion section before final publication in ACP.

**Specific comments:**

- P2, L2 – The authors state "*Our aim is to better quantify particulate emissions from fires over the Amazon basin*". I would argue that the quantification of the emissions is down to

the groups developing these inventories. Rather, the current paper is investigating how different emission datasets affect modelled quantities (e.g. PM, AOT) and evaluate these outputs against a comprehensive set of data collected during the SAMBBA campaign.

- Could you describe the overall methodology behind the GFED, GFAS and FINN products? It would be nice to briefly discuss their strengths and weaknesses which could be used to further support/discuss your modelling results. Were there significant changes between the emissions used in this study (i.e. GFED4, GFAS1.2 and FINN1.5) and those used in Reddington *et al.* 2016 (i.e. GFED3, GFAS1 and FINN1)?

- The Table S1 listing the different optical properties tested is useful and would probably be better located in the main manuscript.

- Are there measurements of aerosol optical properties from the SAMBBA campaign that could be used to further challenge the hypothesis used by GLOMAP in the calculation of AOT?

- Some of the refractive indices listed in Table S1 are derived from Aeronet inversions. Aeronet only provides a bulk column refractive index and cannot artificially separate aerosols into BC and OC components. How do you integrate these values into GLOMAP? Do you apply the same refractive indices for BC and OC when considering the retrieved indices from Aeronet? How does it affect the aerosol absorption regionally? It would be interesting to link that to the different OC/BC ratio from the 3 inventories. Could the simulated Absorption AOT (AAOT) be evaluated against some existing observations then?

- On a similar note, was there any evidence of enhanced absorption from brown carbon during the SAMBBA campaign?

- The sensitivity of AOT to hygroscopic growth constitutes a large uncertainty. It would be useful to show the hygroscopic growth curve response for the two representations considered in this study. The Kolher curve seems to be much more sensitive at higher RH than the GLOMAP parameterisation (e.g. Johnson *et al.*, 2016). Nonetheless, simulated AOT with GLOMAP is much reduced when considering the Kohler model. Could it be due to a lack of representation of subgrid RH in the coarse resolution model? This may be something worth discussing in the model resolution section. In term of meteorological conditions (i.e. RH), was the year 2012 representative of previous years, otherwise could that have an impact on the AOT biases?

- The authors cite the results from Brito *et al.* 2014 indicating that the OA:CO ratios in biomass burning plumes during the SAMBBA campaign suggests limited secondary organic aerosol formation from Amazon fires. The comparison between GLOMAP size distributions and aircraft measurements seems to indicate an underestimation in the Aitken mode. If this is not related to secondary aerosol formation do you have an idea about what causes the discrepancy? Could it be related to the model assumptions on the size distribution parameters and could that have an impact on the calculation of AOT (e.g. moving the accumulation to smaller sizes)?

- It would be good to discuss uncertainties related to the emissions, the measurements and AOT calculation at the end before attributing the modelled AOT underestimation to the way it is calculated. Each measurement technique has its own uncertainties which may vary significantly depending on the observables. In addition, there is additional error that could be related to the sample size and the representativeness of local observations when compared against very coarse model grid-boxes (e.g. Schutgens *et al.*, 2016-2017).

- Please add the definition of NMBF in the main manuscript and explain how to interpretive it. Would the NMBFs listed in Table S3 benefit from being represented graphically to get a better idea of the model skill? (e.g. Figures 3 in Bender *et al.* 2018).

- Page 5, L3 – Figures S2 to S5 are referred before Figure S2 (at L15), reorder.

- Page 5, L29 – Replace Figure 3 by Figure 2. Same at Page6, L9/

- Page 8, L15-17 – "*Sulfate concentrations are well reproduced by the model with no fire emissions and are overestimated when fire emissions are included. This suggests that either emissions of sulfate from fires are overestimated or that other sources of sulfate are overestimated in the model*". I'm struggling to see that from Fig 4b. Is it based on absolute numbers? How does the MACCity compares with say CMIP6 inventories for anthropogenic emissions? It could be nice to show how the different emissions (anthropogenic, dust, BVOCs, …) contribute to the AOT over this region.

- Page 8, L25 – "likely due to a deeper BL over grassland …". Is it confirmed by looking at the model boundary layer height diagnostic? Please define BL acronym also.

- Figure 3 – Use a different symbol in the legend for 'observations' or a different colour so it stands out from the colour used for 'Model (GFED)'.

- Figure 5, L8 – Please state explicitly what STP (i.e. standard temperature & pressure) stands for.

- Figure 8, L9 – change purple with grey in the legend.

- A section on data availability and code availability is necessary to comply with ACP requirements https://www.atmospheric-chemistry-and-physics.net/for_authors/manuscript_preparation.html (see manuscript composition).

**References:**

Bender, F.AM., Frey, L., McCoy, D.T. et al.: Assessment of aerosol–cloud–radiation correlations in satellite observations, climate models and reanalysis, Clim Dyn, https://doi.org/10.1007/s00382-018-4384-z, 2018.

Johnson, B. T., Haywood, J. M., Langridge, J. M., Darbyshire, E., Morgan, W. T., Szpek, K., Brooke, J. K., Marenco, F., Coe, H., Artaxo, P., Longo, K. M., Mulcahy, J. P., Mann, G. W., Dalvi, M., and Bellouin, N.: Evaluation of biomass burning aerosols in the HadGEM3 climate model with observations from the SAMBBA field campaign, Atmos. Chem. Phys., 16, 14657-14685, https://doi.org/10.5194/acp-16-14657-2016, 2016.

Schutgens, N. A. J., Gryspeerdt, E., Weigum, N., Tsyro, S., Goto, D., Schulz, M., and Stier, P.: Will a perfect model agree with perfect observations? The impact of spatial sampling, Atmos. Chem. Phys., 16, 6335-6353, https://doi.org/10.5194/acp-16-6335-2016, 2016.

Schutgens, N., Tsyro, S., Gryspeerdt, E., Goto, D., Weigum, N., Schulz, M., and Stier, P.: On the spatio-temporal representativeness of observations, Atmos. Chem. Phys., 17, 9761-9780, https://doi.org/10.5194/acp-17-9761-2017, 2017.

---

## Referee Comment (RC3) · Anonymous Referee #3 · 12 Oct 2018

The manuscript by Reddington et al. presents an analysis of biomass burning aerosol abundances over the Amazon during the SAMBBA campaign period using a widely used global aerosol model and a collection of observations in order to assess the performance of the model, but also to provide insight into the processes that contribute to common model biases (primarily the underestimation of biomass burning aerosols). Different state-of-the-art emissions datasets are used for driving the simulations so as to explore the sensitivity of model performance to emissions. A discussion of other potential sources of error is also included. The study is a useful contribution to our understanding of why biomass burning aerosols are currently not well captured in modelling. The manuscript is clearly written and certainly within the remit of Atmospheric

Chemistry and Physics. I find the results worthy of publication, following only some minor improvements that I describe below.

SPECIFIC COMMENTS:

Page 2, Lines 27-33: Worth mentioning here which is the wet and which is the dry season, for non-expert readers. Also, in the last sentence of this paragraph it seems that the brackets need to close.

Page 4, Line 1: "using monthly mean 3-D fields at 6-hourly intervals" – what does this mean?

Page 4, Line 5: What about isoprene?

Page 4, Line 7: "Size-resolved emissions of mineral dust are prescribed from daily varying emissions fluxes" – Not clear what this means. Are they dependent on the model's meteorology?

Page 4, Lines 21-25: It is worth giving some additional (brief) information on the fire emissions datasets, e.g. on how they are produced (e.g. based on burnt area or fire radiatve power etc.). Also, any ideas from the literature on why they differ the way they do over different parts of the Amazon (as discussed in the subsequent paragraphs)?

Page 7, Line 13: I realised after finishing reading Section 2 that the start and end date of the simulation as well as the spin-up period have not been mentioned.

Page 7, Lines 22-23: I wonder what it is that made all of them miss the fire emissions just for that part of the season and not for the rest of it. Is it not possible that this could be due to the atmospheric modelling? For example due to the meteorological conditions not being captured well?

Page 8, Line 17: "or that other sources of sulfate are overestimated in the model" – the fact that before the fire season there are no overestimations probably suggests that non-fire emissions are not responsible?
Page 8, Line 25: "and is likely due to a deeper BL over grassland vegetation in the eastern Amazon" – any explanation or reference to support this?

Page 9, Line 12: "in during" -> "during".

Page 11, Line 7: The biases are, however, smaller when comparing with AERONET measurements. Worth mentioning, and perhaps commenting on. What is the uncertainty in observations themselves?

Sect 3.5 (general): Given that this section is a substantial fraction of the manuscript, it would have been nice to support it with some figure or table in the main text. Maybe one simple thing to do is move Table S1 to the main part of the manuscript.

Page 12, Lines 28-29: Size distribution and composition are not discussed substantially in this analysis of possible factors. Is there anything more that could be said about them? Also, what about meteorological quantities other than humidity, e.g. wind or temperature?

Page 13, Lines 18-19: "but not fully resolve the negative bias in model AOD" – please add "...which is of the order of XX%" to give a sense of how far the results would still be.

---

## Author Comment (AC1) · 4 Apr 2019

We would like to thank the referees for taking time to review our manuscript and for all the comments they have provided. We have responded to all the referee comments below and have modified our manuscript accordingly. Our manuscript has been strongly improved through this process and we hope it is now suitable for publication.

Other changes that we have made during the review process are as follows:
- Correction of minor errors/typos.
- Alteration of the size of the western Amazon region to be more in-line with Darbyshire et al. (2018), which was submitted to ACPD after this manuscript. This alteration had minor effects on values/figures in the manuscript (please see the revised manuscript) and, importantly, did not have any effects on the conclusions of our paper.
- Improved temporal co-location of simulated data with aircraft measurements; removing simulated data corresponding to invalid measurement points. This resulted in minor changes to Fig. 5 and bias values in Table 2.
- Addition of Catherine Scott as a co-author for assistance with responding to reviewing comments regarding biogenic secondary organic aerosol.

To guide the review process, referee comments below are in plain text and our responses are in italics, additions to our manuscript are shown below in red and as highlighted sections in the revised manuscript.

Please find responses to individual referees on the following pages:

Page 2 – Response to Anonymous Referee #1

Page 12 – Response to Anonymous Referee #2

Page 19 – Response to Anonymous Referee #3

**Anonymous Referee #1**

This study uses a combination of modeling (with various emission inventories) and observations to explore the importance of biomass burning aerosol over the Amazon. The study does a nice job of bringing together different observations and the comparisons against various inventories are enlightening (if only to understand how very uncertain biomass burning emissions are!). There are a few areas that should be improved to enhance the robustness of the study and its utility to the community before final publication:

*Thank you for the positive comments on our study.*

1. The authors discuss a number of factors that could influence their comparison with observations, but disappointingly, they don't perform any sensitivity simulations to test these. It would be nice to see the impact of (a) injecting all fire emissions at the surface and (b) decreasing the size of emitted biomass burning aerosol in the model on their results. This would enable the authors to be conclusive about these factors; without this the discussion remains largely unsubstantiated.

   *We have performed two additional sensitivity simulations as suggested:*

   *Injecting all fire emissions at the surface results in relatively little difference in the vertical profile of aerosol (see Fig. S11), demonstrating that vertical mixing rapidly redistributes aerosol, particularly in the western region. Simulated September-mean surface organic aerosol (OA) mass concentrations increase by ~88% in the eastern region and by ~15% in the western region, relative to the control simulation using injection heights specified by AEROCOM, but decline relative to the control above ~1 km in the east and above ~350 m in the west, so mean changes in the boundary layer are small (+5% in the east < 4 km; -0.3% in the west < 2.5 km). We have added an additional figure to the supplementary (Fig. S11) and the following text to the revised manuscript (Sect. 3.2):*

   "Figure S11 shows average aerosol mass vertical profiles from a simulation with all GFED emissions injected into the model surface layer ("GFED_surflev"), as opposed to vertically distributed as described in Sect. 2.2.1. Injecting fire emissions into the surface layer has a relatively small impact on the simulated aerosol vertical profile in the west (a mean change of -0.3% in OA mass below 2.5 km), with a small change in the bias against aircraft observations (e.g. for OA in Phase 1; NMBF= -0.82 for GFED and -0.77 for GFED_surflev), demonstrating that vertical mixing rapidly redistributes aerosol in the model. In the eastern region, the impact on simulated aerosol mass is larger (a mean change of +5% in OA mass below 4 km), reducing the negative model bias (e.g. for OA; NMBF= -1.23 for GFED and -0.90 for GFED_surflev)."

   *Altering the assumed emission size distribution to better match size distribution measurements of biomass burning aerosol (e.g. Reid et al., 2005) increased the simulated number concentration of particles < ~100 nm (Fig. S12) but reduces AOD (mean changes of -13% in the west and -9% in the east). We have added a figure to the supplementary comparing the two emission-size sensitivity simulations to aircraft measurements of the number size distribution (Fig. S12) and we have added the following text to Sect. 3.3:*

   "We performed two sensitivity tests where we varied the assumed emission size distribution for primary biomass burning aerosol in GLOMAP (see Fig. S12). Reid and Hobbs (1998) measured count median diameters (CMD) of 130±10 nm (σ=1.68±0.02) and 100±10 nm (σ

=1.77±0.02) for deforestation fires and 100±10 nm (σ=1.91±0.15) for Cerrado fires (Reid et al., 2005). Assuming a CMD of 100 nm increases the simulated particle number concentration below 100 nm diameter by factors of ~1.8 (with σ=1.7) and ~1.5 (with σ=1.8) over the SAMBBA regions. This results in a reduction in the negative bias in simulated number concentration above 50 nm (N50; GFED (150 nm, σ=1.59): $NMBF_{WestP1}$=-1.85; GFED (100 nm, σ=1.7): $NMBF_{WestP1}$=-0.51), but a slight increase in the negative bias in N200 (GFED (100 nm, σ=1.7): $NMBF_{WestP1}$=-0.55). Therefore, reducing the assumed emission size distribution for primary BC and OC from biomass burning may be important for cloud condensation nuclei concentrations, but will have a small effect on simulated aerosol mass and AOD (see Sect. 3.5.1)."

*We also test the sensitivity of simulated AOD to reducing the assuming emission size and injection height of biomass burning aerosol. We have included an additional section (Sect. 3.5.1 in the revised manuscript) and figures (Fig. 9 and Fig. S14) summarising these results. We have also added additional text to the conclusions section regarding these sensitivity simulations.*

2. Observational uncertainties. The authors should include some information on the uncertainties associated with the various measurement techniques in Section 2.3 and include these uncertainties in their discussion of the measurement-model comparisons. In particular, aethalometer observations are highly uncertain with significant filter loading artifacts. The authors should acknowledge this and discuss what impact it might have on their results.

*We have now added sections to the main paper (Sect. 2.3.5 in the revised manuscript) and supplementary material (Sect. S2) describing further details of the instrumentation used during SAMBBA; including information about measurement calibration and uncertainty. We have also added a new section (Sect. 3.6 in the revised manuscript) that includes a discussion of measurement uncertainty and its impact on the AOD comparisons. In general we find that measurement uncertainty (~15-30%) is smaller than uncertainty in the calculation of AOD (e.g. ~40% for water uptake) and model representation of relative humidity variability (up to ~ 100%). We have added the following text to the revised manuscript:*

*Sect 2.3.5:*
"Section S2 describes further details of the instrumentation used during SAMBBA; including information about measurement calibration and uncertainty. In summary, for conditions during SAMBBA the mass concentration measurement uncertainty has been estimated to be: ~20% for the aethelometer (Schmid et al., 2006); 10-35% for the ACSM (depending on the species, OA is 15%; Crenn et al., 2015); ~30% for the AMS (Bahreini et al., 2009; Middlebrook et al., 2012); and ~30% for the SP2 (Schwarz et al., 2008; Shiraiwa et al., 2008). For AOD retrievals, the 1σ uncertainty is estimated to be ±0.05+0.15% for MODIS (Levy et al., 2010) and ±0.01 AERONET (Giles et al., 2019)."

*Sect. 3.6:*
 "The uncertainty in the MODIS retrievals of AOD is important to consider (about ±30% for the magnitude of AOD observed during the SAMBBA campaign; Sects. 2.3.5 and S2), but is considerably smaller than the uncertainties associated with simulated biomass burning aerosol properties and AOD."

3. Are there sufficient number of observations in the "eastern region" to be statistically representative? It is hard to see from Figure 2 how many flights extend over this region and it might be useful to include the number of observations per vertical bin in Figure 5. The authors should acknowledge the limited sampling here.

*We believe that the amount of measurement sampling in the eastern region is sufficient to draw conclusions from (as in e.g., Archer-Nicholls et al. (2015) and Hodgson et al. (2018)), but we agree that the aircraft sampling in this region is limited relative to sampling in the western region and should be acknowledged. We have added a figure to the supplementary material (Fig. S8) showing the number of AMS and SP2 measurements per vertical bin in Fig. 5 for the eastern and western regions. We have also added the following text to the manuscript:*

*To Section 2.3.1:* "We note that the aircraft sampling in the eastern region (including one full flight and sections of three flights) was limited relative to the sampling performed in the western region (including 14 full flights and sections of five flights)."

*To the beginning of Section 3.2:* "We note that the aircraft sampling in the eastern region was limited relative to sampling in the western region (Sect. 2.3.1). Figure S8 shows the number of OA (from the AMS) and BC (from the SP2) observations per vertical bin for the western region (Phases 1 and 2) and eastern region."

*To the conclusions:* "This suggests that all emission datasets may underestimate aerosol emissions from grassland/savannah fires in the eastern Amazon, although we acknowledge the limited measurement sampling in this region relative to the western Amazon."

4. Section 3.2: The authors show that the model fails to capture the decrease in aerosol concentrations from Phase 1 to Phase 2. Did the authors explore what role, if any, meteorology might play (temperature, wind direction, precipitation)? The emission inventory may not adequately reflect changes in burn conditions.

*Thank you for this suggestion. We agree that the model meteorology might be partly responsible for the failure of the model to capture the decrease in aerosol concentrations from Phase 1 to Phase 2 (particularly in the vertical profile of aerosol mass). We have added the following text to the revised manuscript in Sect 3.2:*

"This may be because the emission datasets report only moderately lower emissions in Phase 2 compared to Phase 1 (Figure S3a; Table 1), but also because the model may underestimate wet removal of aerosol during Phase 2 (consistent with model and observation comparisons in Archer-Nicholls et al. 2015)."

*To give more detail, the observed organic aerosol mass decreases between Phase 1 and Phase 2 by a factor of ~3.4 at the surface at Porto Velho and a factor of ~3.2 below 2.5 km altitude. As discussed in the paper, the model simulates a smaller reduction in organic aerosol mass concentrations between Phase 1 and 2 than observed (a factor 1.5-1.7 at Porto Velho; a factor ~1.1-1.2 below 2.5 km altitude); the simulations with FINN and GFED emissions yielding the smallest and largest reductions, respectively. In the fire emissions datasets, the total organic carbon emissions decrease by a factor of ~1.9-3.4 between Phase 1 and Phase2 (smallest decrease in FINN; largest in GFED). The changes in emissions are generally smaller than observed changes in organic aerosol mass concentrations, but the observations likely reflect reductions in fire emissions combined with an increase in wet removal. Therefore, it is possible that the model underestimates wet removal of aerosol in Phase 2.*

5. Figures 7 & 8, and page 11, lines 3-11: The authors suggest that Figures 7 and 8 are consistent, but they do not appear to be so. Figure 7 shows a clear underestimate in mean MODIS AOD over the western and eastern Amazon by all of the models, by a factor of ~2. Figure 8 shows that at least some of the models adequately capture (and sometimes overestimate!) the AOD observed at AERONET sites and by MODIS at these sites, with the exception of the early part of the Alta Floresta record. Therefore, the statement on page 11, line 7 "the model consistently underestimates..." is clearly false. Given the reasonable agreement between AERONET and MODIS at these sites, does this analysis suggest that the MODIS AOD observed in other regions of the Amazon is biased high? The authors need to correct their conclusions and discuss this more fully.

*We have now adjusted the size of the region in the west that we average the MODIS data over, to be more in-line with Darbyshire et al. (2018) and to be more consistent with the region impacted by deforestation fires (in the previous version of the manuscript we were averaging MODIS and model AOD550 over large sections of pristine forest). The model biases in AOD500 (against AERONET) and AOD550 (against MODIS) are now more consistent:*

|                          | noBBA | FINN  | GFED  | GFAS  |
|--------------------------|-------|-------|-------|-------|
| Western Amazon, Phase 1  |       |       |       |       |
| AOD550 (MODIS)           | -5.25 | **-0.51** | -0.98 | -1.43 |
| AOD500 (AERONET)         | -6.95 | **-0.47** | -0.53 | -1.26 |
| Western Amazon, Phase 2  |       |       |       |       |
| AOD550 (MODIS)           | -3.68 | **-0.38** | -0.70 | -1.06 |
| AOD500 (AERONET)         | -4.50 | **-0.41** | -0.68 | -1.11 |

*If we reduce the averaging region for MODIS AOD further (to fit more tightly around the western AERONET stations: 54-68.5°W, 7-12°S) and compare model biases in AOD550 over this region to model biases in AOD500 only at the three AERONET stations in the western region, again we see that they are consistent (e.g., with GFED emissions: AERONET AOD500, $NMBF_{P1}$=-0.58, $NMBF_{P2}$=-0.49; MODIS AOD550, $NMBF_{P1}$=-0.64, $NMBF_{P2}$=-0.44).*

*We note that caution should be taken when quantitatively comparing AOD550 (from MODIS) and AOD500 (from the model and AERONET) in Fig. 7 due to the differences in the wavelengths. In the model, where we can compare AOD calculated at both wavelengths, AOD500 is higher than AOD550. In addition, the data coverages of MODIS (at the specific AERONET station locations) and AERONET AOD is different, and so the analysis period would need to be extended to look in more detail. Therefore, we cannot ascertain from this analysis that MODIS AOD observed in other regions of the Amazon is biased high. We suggest the differences in model agreement with MODIS AOD550 and AERONET AOD500 may be due to differences in e.g., AOD wavelengths, AERONET/MODIS measurement uncertainties, location/region of comparison, and data coverages.*

*With regard to the statement on page 11, line 7, the bias across all AERONET stations (see table above) does demonstrate that the model underestimates AOD500 at these locations. However, we agree that AOD500 is not consistently underestimated at all AERONET stations during the campaign. Therefore, we have made the following changes and additions to the text in Sect. 3.4:*

"Consistent with comparisons to MODIS, the model generally underestimates AOD500 at all stations and with all fire emissions, except at two stations in the western Amazon (Rio Branco (in both campaign phases) and Porto Velho (in Phase 2)) with FINN emissions. The negative model bias in AOD500 across all AERONET stations is consistent with the negative model bias in AOD550 (against MODIS) (Table 2), but is smaller at some individual stations (Fig. 8). This is likely due to multiple reasons including differences in: i) the AOD wavelengths (500 nm versus 550 nm); ii) the AERONET and MODIS retrieval uncertainties (Sect. S2.3); iii) the location/region of comparison, affecting magnitude and sources of AOD; and iv) the AERONET and MODIS data coverages."

*We have also changed "consistently" to "generally" throughout the paper, including Abstract and Conclusions.*

6. Section 3.5.3: The authors make a clear case that uncertainties associated with water uptake have a large impact on simulated AOD, but appear to show that this cannot explain the discrepancy between observations and their model (if they use an alternate approach the bias gets worse).  Thus, as I read this, the authors fail to come up with any explanation for why simulated AOD appears biased low when mass concentrations are captured by their model. If so, the authors should be clear that this is not resolved and modify their conclusions and abstract accordingly.  If they are saying that the uncertainty associated with water uptake could increase the AOD, they should show this result.

*We agree that we could have stated more clearly that the low bias of model AOD was not resolved in our analysis. The aim of Section 3.5 is not to solve the discrepancy in model AOD in this region but to highlight that AOD depends on many uncertain variables and so to caution against scaling model biomass burning aerosol emissions or concentrations to match AOD retrievals. We have reworded the manuscript to make this clearer. We have added additional discussion (Sect. 3.6) and figures of the AOD sensitivity studies (Fig. 9 and Fig. S14).*

**Other Minor Comments/Corrections**

1. Page 2, Line 2:  It seems that the goal of this study is to quantify the impacts of biomass burning emissions, not the emissions themselves (i.e.  this isn't an inverse modeling study and the authors did not present a best-estimate of emissions from fires in the region). Please re-phrase.

*We agree. We have re-phased this sentence to the following:*

"Our aims are to: 1) quantify the effects of biomass burning emissions on the aerosol distribution over the Amazon; and 2) explore how different fire emissions datasets affect simulated aerosol concentrations over this region."

2. Section 2.1:  The model description is missing a few items:  (a) the year of meteorology, (b) a description of aerosol removal (wet & dry), and (c) production totals of biogenic SOA for the region to compare to Table 1 emissions (also: did the simulation not include isoprene SOA?? This seems like an oversight).

*The year of meteorology used matches the simulation date/time. We have added the following text to Sect 2.1 to make this clearer:*

"Simulations were run from 1[st] January 2003 to 31[st] December 2012, using ECMWF ERA-Interim reanalyses that correspond to the simulation date/time."

*Detailed descriptions of the dry and wet aerosol removal can be found in Mann et al. (2010). We have added the following text (also to Sect. 2.1) to give more information:*

"Below we describe the features of the model relevant for this study, please see Spracklen et al. (2005) and Mann et al. (2010) for more detailed descriptions of the model and see Reddington et al. (2016) for further details of the model set-up used here."

"Wet removal of aerosol in GLOMAP occurs by two processes: 1) in-cloud nucleation scavenging, calculated for both large-scale and convective-scale precipitation based on rain-rates diagnosed from successive ECMWF ERA-Interim reanalysis fields; and 2) below-cloud impaction scavenging via collection by falling raindrops. For dry deposition of aerosol, GLOMAP calculates the wind speed and size-dependent deposition velocity due to Brownian diffusion, impaction and interception. Detailed descriptions of the dry and wet aerosol removal process are in Mann et al. (2010)."

*We are unable to provide production totals of biogenic SOA for the region for this time period because in the set-up of the model used in this study, the SOA component is not tracked individually (but is combined with primary organic carbon aerosol in the particulate organic matter component). The global production rates of SOA in GLOMAP have been estimated in Scott et al. (2014) (see Table 2), by combining the emissions biogenic volatile organic compounds with the assumed production yield of SOA, but regional production rates cannot be calculated in this way.*

*We do not include isoprene SOA in this model set-up in line with recent evidence showing isoprene does not lead to net production of SOA mass (McFiggans et al., 2019). In a different version of the GLOMAP model (see Scott et al. (2018)), SOA from both monoterpenes and isoprene are included. From the model version used in Scott et al. (2018) we find that isoprene SOA contributes ~0.5-2 µg m[-3] and monoterpene SOA contributes ~0.5-5 µg m[-3]to the September-mean total organic aerosol (OA) mass concentration over the Amazon region (see figures below).*

[Figure]

[Figure]

[Figure]

*We also note that in Scott et al. (2018) the yields of both isoprene SOA and monoterpene SOA (which are both highly uncertain) had to be scaled down in order for simulated SOA to match observed organic carbon mass in the wet season in the Amazon. Comparing with the same measurements as used in Scott et al. (2018) (in their supplementary material), we find that the GLOMAP model version used in this study captures the magnitude of measured organic carbon (OC) mass concentration in the wet season without isoprene SOA (see figure below). Which suggests that simulated SOA mass concentrations in this study are reasonable and broadly consistent with the findings of McFiggans et al. (2019).*

[Figure]

3.  Section 2.2:  The authors focus their comparison of inventories on the OC emissions.  They should comment somewhere in the text on whether the differences in BC emission (shown in Figure 1) are similar.

*The spatial patterns in emissions (and in the differences between 2012 and 2002-2012) are very similar for OC and BC emissions. We have added the following text to the revised manuscript:*
"Total annual BC emissions show very similar spatial patterns to the OC emissions shown in Figs. 1 and S1."
"Figure 1 also shows the difference in annual total OC emissions between 2012 and the 2002-2012 mean (very similar spatial patterns are seen for BC emissions)."

4. Page 5, line 29 & page 6, line 9: typo: Figure 2

*Thank you. Now corrected.*

5. Page 6, lines 12-13: briefly describe the main features of the plume removal algorithm, and how much data was removed using this filtering.

*For each time series this comprised defining a baseline (moving 5-minute $25^{th}$ percentile), an upper threshold for plume identification (baseline + campaign $90^{th}$ percentile) and a lower threshold to determine the plume extent (baseline + campaign $10^{th}$ percentile). If any of the identifier species (CO, $CO_2$, refractive BC mass or aerosol scattering) passed the threshold a plume was defined. Of all SLR data collected during the campaign such plumes comprised 10%.*

6. Section 2.3.4: include more information on the retrieval and relevant product including appropriate references.

*We have now included more information about the MODIS aerosol optical depth product that we used in our analyses. The following text has been added to Sect. 2.3.4:*
"Specifically, we used the Collection 5.1 Level-3 MODIS Atmosphere Daily Global Product gridded to 1°×1° resolution (Terra: MOD08_D3; Aqua: MYD08_D3; https://modis-atmosphere.gsfc.nasa.gov/products/daily) (Hubanks et al., 2008) acquired through NASA's Level 1 and Atmosphere Archive and Distribution System (LAADS) (https://ladsweb.modaps.eosdis.nasa.gov/)."

7. Page 7, lines 28-30:  This text is confusing.  Figure 4a shows maximum concentrations of 100 ug/m3 (not 30-40 ug/m3) – did the authors mean to refer to Figure 3 here? Also the ACMS+BCeq in Figure 4 is not "consistently lower" than PM2.5 in Figure 3 as it is clearly higher on Sep 14 and 22. Please correct this text.

*Thank you for pointing this out. We agree the text is confusing and contains a mistake (we meant to refer to Fig. S6 rather than Fig. 4a). We have modified the text to the following:*

"Measured total aerosol mass, calculated as mass measured by the ACSM plus $BC_{eq}$ measured by the aethelometer, varies consistently with measured PM2.5 concentrations during the campaign (Fig. S6). However, when averaged over the gravimetric filter analysis sampling time, measured total (ACSM+$BC_{eq}$) aerosol mass concentrations are consistently lower than measured PM2.5 concentrations by ~20-60% (Fig. S6a)."

8. Page 8, lines 1-3:  might differences in measurement technique (beyond size cut-offs) also be a factor?
*Yes, it is certainly possible that this may play a role in the differences in measured aerosol mass concentration. We suggested this by referring to possible aerosol species that are not detected by the ACSM, but we agree that this could have been made clearer. We have changed the sentence to the following:*
"This difference in the measurements is mostly apportioned to the reduced aerosol detection-size range from the ACSM (i.e. submicrometric) in comparison to the gravimetric analysis (< 2.5 µm) (Sect. 2.3.2), and, to a smaller extent, the different measurement techniques e.g. aerosol species unaccounted by the on-line instrumentation (ACSM) e.g. crustal elements."

9. Page 8, lines 13-15: how much of simulated OA is biogenic SOA?

*Please see response to "Minor comment 2" above. Using a different version of the model (used in Scott et al. (2018)), we estimate that monoterpene SOA contributes ~0.5-5 µg m$^{-3}$ to the September-mean total organic aerosol (OA) mass concentration over the Amazon region. For context, this equates to ~3-30% of the campaign-mean simulated surface OA mass concentration at Porto Velho (for the model simulation with GFED4 fire emissions).*

10. Page 8, line 14: is the NH4+NO3+Chl contribution relatively consistent in the observations or are there days where these species make a larger contribution to total measured PM2.5?

*The contribution of NH4+NO3+Chl is relatively consistent in the observations during the SAMBBA campaign, varying between~1 and 12% of the total aerosol mass (median fraction of total mass: 6.1%; 25$^{th}$ to 75$^{th}$ percentile range: 4.4% to 7.7%).*

11. Page 8, lines 13-15: how well does GLOMAP capture the speciated mass concentrations? i.e. what is the $R^2$ between simulated and observed OA and BC?

*GLOMAP captures the magnitude and variability of hourly mean measured OA and BC concentrations during SAMBBA reasonably well. For example, for the simulation with GFED fire emissions, OA: NMBF= 0.3, $R^2$ = 0.4; BC: NMBF=-0.4, $R^2$ = 0.4. We have now included a figure in the supplementary material showing the hourly time-series of simulated and observed BC and OA at Porto Velho (see new Fig. S7).*

12. Figure 5, 6, 7, 8: would be more legible if the authors included a legend.

*We agree. Legends have now been added to Figures 5, 6, 7, 8.*

13. Page 8, line 25: Is the boundary layer deeper in the eastern Amazon than western Amazon in the model? Please comment on this in the text.

*We have now included a figure in the supplementary material (Figure S10) showing the September 2012 mean boundary layer height (calculated from daily mean boundary layer height from ECMWF) over the Amazon and wider region. This figure demonstrates that the boundary layer height in the eastern region was generally greater than in the western region in September 2012. We now refer to this figure in the text (Page 12, line 9 in the revised manuscript).*

14. Page 9, line 12: typo "~0.5 ug/m3 during Phase 2".

*Thank you. Now corrected.*

15. Page 9, lines 24-29: Do the differences in observed BC:OA mass concentrations indicate anything about differences in fuel type or burn conditions in the western vs. eastern Amazon?

*Yes. Measurements in the eastern region are representative of biomass burning aerosol from small, Cerrado (savannah/grassland type) flaming fires that occurred in Tocantins state; while measurements in the western region are likely more representative of larger, smouldering tropical forest fires that occurred in and around Rondônia state (Hodgson et al., 2018). We have now modified the sentence referenced in the comment to the following:*

"These ratios reflect the much higher BC emission factors found for flaming Cerrado fires in the eastern Amazon relative to tropical forest fires in the western Amazon (Hodgson et al., 2018)."

16. Page 10, line 11: why use only "straight and level runs"? How many measurements are included in these averages?

*The SMPS measurements are not provided for the aircraft vertical profiles because they do not provide reliable information throughout them, due to the instrument sampling time (a few minutes) (this explanation is now included in the new Sect. S2). Therefore, we only use SMPS measurements from straight and level runs, for which the data is reliable. The number of valid measurements in each altitude bin depends on the size bin of interest. For the 103 nm size bin in Figure 6, we averaged over the following number of valid data points:*

| SAMBBA period | 0-2000 m altitude bin | 2000-4000 m altitude bin |
|---|---|---|
| Western region, Period 1 | 8448 | 2369 |
| Western region, Period 2 | 23988 | 7320 |
| Eastern region | 2689 | 98 |

17. Page 13, lines 9-12: Have the authors compared the highest RH? These are the values that will disproportionately impact water uptake and aerosol growth. How well does the model capture observed RH > 90%?

*Thank you for this suggestion. We have now improved the figure showing the comparison of model and observed RH vertical profiles (see Fig. S15) so that observed RH values > 90% are visible. Figure S15 shows that the model does not capture observed RH > 90%. There are relatively few observed RH values above 90%, particularly where aerosol concentrations are highest (within the BL), but we agree these high RH values could be important for AOD. To examine this further we carried out additional sensitivity tests where we set the model RH to the mean or maximum observed RH in each model level (on days with available aircraft data) and calculated the resulting water uptake using the κ-Köhler scheme. We have included the results of these sensitivity tests in Sect. 3.5 and in Figs. 9, 10 and S14.*

**Anonymous Referee #2**

**Summary:**

Biomass burning is a major source of particulate matter pollution, regionally and globally. This has important implications for air quality and climate. Over South America, intense fires occur in August-September typically, providing the dominating source of atmospheric aerosols to the region. Yet considerable uncertainties in the magnitude of fire emissions remain. As such, the paper focuses on 'improving understanding of aerosol emissions from vegetation fires' by considering three different fire emission datasets (namely GFED4.1, GFAS1.2 and FINN1.5) to account for fire emissions in their modelling work. The authors used a global aerosol model (GLOMAP) to study how the simulated particulate matter (PM) concentration and aerosol optical thickness (AOT) are affected by the three different fire emission inventories. These results are compared against a comprehensive set of surface, aircraft and satellite observations collected over the Amazon region during September 2012. The authors have highlighted the spatial and temporal variation in the three different fire emissions and how it affects simulated quantities. Overall, the authors conclude that GLOMAP has skill in predicting reasonable surface concentration and vertical profile of PM over South America despite noticeable differences between the emission inventories. However, GLOMAP simulated AOT is found to be systematically underestimated. The authors therefore recommend caution when evaluating global models using AOTs to constrain particulate emissions from fires.

**General comments:**

The work presented here shares a lot of its DNA with a previous publication from the same group (i.e. Reddington et al., 2016). In this previous effort, the authors used the same modelling framework to argue that GLOMAP showed better agreement with observed PM mass concentration compared to AOT, potentially suggesting that some of the discrepancy between top-down and bottom-up studies may be connected to the calculation of AOT. In the present work, the authors test this hypothesis further by: *i*) providing a much more detailed evaluation of the GLOMAP model simulations against a comprehensive set of observations collected over South America during the SAMBBA campaign, and *ii*) performing a model sensitivity analysis exploring the assumptions related to the calculation of AOT in GLOMAP.

The paper is well structured and reads easily. The model evaluation is rigorous and convincing. The figures are clear and illustrate the points made in the manuscript. This paper is interesting and has a good potential. The last section of the result (Sect. 3.5) however feels too rushed in its current form and could benefit from the support of more visual material (i.e. show some plots for these results). The results in this section are mentioned too briefly, and do not provide a critical interpretation that would ensure more generally applicable results that could be subsequently transferred to other atmospheric aerosol models. The paper is quite weighted towards model evaluation. A model evaluation is only really useful if it used to interpret observed relationships or processes. As such, section 3.5 fails to convincingly demonstrate the assumption tested in this study.

The diversity in fire emissions highlighted here has important implication for aerosol modelling over this region, and likely in any region influenced by biomass burning. Contrasting the uncertainties from emissions with the uncertainties related to AOT calculation could really improve the scientific strength

of the work. I would suggest reworking section 3.5, and perhaps add a discussion section before final publication in ACP.

*Thank you for the positive comments about the manuscript.  As suggested, we have added new figures (Figures 9, 10 and S14) to summarise the results from the sensitivity studies and we have added additional sensitivity studies and discussion to Sect. 3.5 (and a new section; Sect 3.6). Figure 9 demonstrates that uncertainties associated with the AOD calculation result in larger differences in simulated AOD compared to differences in emission datasets. This confirms the issues involved with using AOD to help constrain biomass burning emissions.*

**Specific comments:**

1. P2, L2 – The authors state "*Our aim is to better quantify particulate emissions from fires over the Amazon basin*". I would argue that the quantification of the emissions is down to the groups developing these inventories. Rather, the current paper is investigating how different emission datasets affect modelled quantities (e.g. PM, AOT) and evaluate these outputs against a comprehensive set of data collected during the SAMBBA campaign.

   *We agree and thank the reviewer for the suggested revision. We have re-phrased these sentences to the following:*

   "Here we evaluate the Global Model of Aerosol Processes (GLOMAP; Spracklen et al., 2005) against a comprehensive set of measurement data (including surface, aircraft and satellite observations) collected during the South American Biomass Burning Analysis (SAMBBA) field campaign in September/October 2012 over the Amazon basin. Our aims are to: 1) quantify the effects of biomass burning emissions on the aerosol distribution over the Amazon; and 2) explore how different fire emissions datasets affect simulated aerosol concentrations over this region."

2. Could you describe the overall methodology behind the GFED, GFAS and FINN products? It would be nice to briefly discuss their strengths and weaknesses which could be used to further support/discuss your modelling results. Were there significant changes between the emissions used in this study (i.e. GFED4, GFAS1.2 and FINN1.5) and those used in Reddington *et al.* 2016 (i.e. GFED3, GFAS1 and FINN1)?

   *We have now included a new section describing the fire emissions datasets and including information on updates that have been documented (please see Sect. 2.2.2 in the revised manuscript). We are unable to compare GFED3 and GFED4 emissions for 2012 as GFED3 emissions are not available for this year. The differences in total annual emissions of organic carbon over the region shown in Fig. 1 are fairly small between FINN1.0 and FINN1.5 (~4%) and between GFAS1.0 and GFAS1.2 (~1%) on a 0.5°x0.5° grid.*

3. The Table S1 listing the different optical properties tested is useful and would probably be better located in the main manuscript.

   *Thank you for the suggestion. Table S1 has now been moved into the main manuscript (as Table 3) and extended to include more sensitivity tests.*

4. Are there measurements of aerosol optical properties from the SAMBBA campaign that could be used to further challenge the hypothesis used by GLOMAP in the calculation of AOT

*Yes, we have now incorporated aircraft measurements of aerosol extinction and scattering coefficients into our analysis (see Table 2 and Sects. 3.4 and 3.5) and included an additional figure in the supplementary (Fig. S13).*

5. Some of the refractive indices listed in Table S1 are derived from Aeronet inversions. Aeronet only provides a bulk column refractive index and cannot artificially separate aerosols into BC and OC components. How do you integrate these values into GLOMAP? Do you apply the same refractive indices for BC and OC when considering the retrieved indices from Aeronet? How does it affect the aerosol absorption regionally? It would be interesting to link that to the different OC/BC ratio from the 3 inventories. Could the simulated Absorption AOT (AAOT) be evaluated against some existing observations then?

*Yes, we apply the same refractive indices to the BC and POM components in the model when testing the AERONET-derived refractive indices. We agree that it would be interesting the look at how this affects the aerosol absorption regionally, but to evaluate the model absorption in detail we would need to extend the analysis period to acquire sufficient AAOT data from the AERONET stations and is beyond the scope of this current study.*

*In our (newly added) comparisons to the aircraft vertical profiles of extinction and scattering coefficients, it is evident that there are competing impacts of applying the refractive indices in this way on the simulated aerosol extinction and scattering. For example, applying the refractive indices retrieved by Ndola AERONET station (16 Sep 2000) reduces the negative bias in model aerosol scattering against the aircraft measurements, but increases the negative model bias in aerosol extinction. On the other hand, setting the BC-component refractive indices to the mid-range value for light absorbing carbon, acts to increase the negative bias in model aerosol scattering against the aircraft measurements, but reduce the negative model bias in aerosol extinction. We stress, however, that these effects are relatively small.*

6. On a similar note, was there any evidence of enhanced absorption from brown carbon during the SAMBBA campaign?

*Measurements of brown carbon have been performed at the Amazon Tall Tower Observatory (ATTO; Saturno et al., 2018), but as far as we are aware there has been no specific analysis of absorption from brown carbon during the SAMBBA campaign. Brown carbon is not a focus of this study, so we point the reviewer to the paper by Saturno et al. (2018) for more information.*

7. The sensitivity of AOT to hygroscopic growth constitutes a large uncertainty. It would be useful to show the hygroscopic growth curve response for the two representations considered in this study. The Kolher curve seems to be much more sensitive at higher RH than the GLOMAP parameterisation (e.g. Johnson *et al.*, 2016). Nonetheless, simulated AOT with GLOMAP is much reduced when considering the Kohler model. Could it be due to a lack of representation of subgrid RH in the coarse resolution model? This may be something worth discussing in the model resolution section. In term of meteorological conditions (i.e. RH), was the year 2012 representative of previous years, otherwise could that have an impact on the AOT biases?

*Thank you for raising this point. As the reviewer mentions, the Kolher curve is more sensitive at higher RH than the GLOMAP parameterisation as shown in Johnson et al. (2016). The flattening off of the curve in Fig. 13 of Johnson et al. (2016) is because the RH is restricted in*

*GLOMAP to between 10 and 90% when using ZSR to calculate water uptake. The steep Kohler curve at higher RH (and the restriction of RH for ZSR), however, are not having an impact on the calculated growth factor in this analysis because the model RH does not go above 90% for the SAMBBA analysis period and regions.*

*We have now improved the figure evaluating model RH against the RH measured by the aircraft (now Fig. S15), which reveals the model underestimates the variability in observed RH. Therefore, we agree that inadequate representation of sub-grid RH may be one cause of the model discrepancy in AOD.*

*We have added additional sensitivity tests to Sect. 3.5 where we set the model RH to the mean or maximum observed RH in each model level (on days with available aircraft data) and calculated the resulting water uptake using the κ-Köhler scheme (described in new section Sect. 3.5.5). These simulations have mixed results on simulated AOD (summarised in new Figs. 9, 10 and S14). Setting the model RH to the mean observed RH results in a small increase in simulated AOD in the east (by ~11%) and small decrease in the west (by ~9%) (relative to model AOD calculated using κ-Köhler and GLOMAP RH). Setting the model RH to the maximum observed RH increases model AOD (by~75-100%) improving agreement with MODIS, but leads to overestimation of the aerosol scattering and extinction coefficients between 4 and 6 km altitude (see Sect 3.5.5 in the revised manuscript).*

8. The authors cite the results from Brito *et al.* 2014 indicating that the OA:CO ratios in biomass burning plumes during the SAMBBA campaign suggests limited secondary organic aerosol formation from Amazon fires. The comparison between GLOMAP size distributions and aircraft measurements seems to indicate an underestimation in the Aitken mode. If this is not related to secondary aerosol formation do you have an idea about what causes the discrepancy? Could it be related to the model assumptions on the size distribution parameters and could that have an impact on the calculation of AOT (e.g. moving the accumulation to smaller sizes)?

*We agree that the particles in the nucleation/Aitken modes are likely related to SOA formation. The evidence from (Brito et al., 2014; Morgan et al., 2019) suggests little net change to total organic aerosol mass from in-plume SOA formation, but there may be a contribution to particle number. In the paper we say:*

"The observations suggest biomass burning makes a considerable contribution to aerosol number from ~50 to 200 nm diameter that is not included in the model. This is consistent with Vakkari et al. (2018), where assumed emission size distributions in models poorly represented the number of particles in the 30–100 nm (Aitken mode) size range for southern African savannah and grassland fires. The particle number in the Aitken mode size range will have a negligible effect on AOD but may be important for cloud condensation nuclei concentrations."

*To improve simulation of particle number in the smaller mode, it is likely that we need to either add in-plume formation of SOA for biomass burning in the model (or assume a bimodal emission size distribution for biomass burning emissions). Reducing the assumed emission for biomass burning emissions (from a count median diameter of 150 nm to 100 nm) improves simulation of the number of smaller particles but increases the negative bias of accumulation-mode particle number concentration and AOD (see Sect. 3.3 of the revised manuscript).*

9. It would be good to discuss uncertainties related to the emissions, the measurements and AOT calculation at the end before attributing the modelled AOT underestimation to the way it is calculated. Each measurement technique has its own uncertainties which may vary significantly depending on the observables. In addition, there is additional error that could be related to the sample size and the representativeness of local observations when compared against very coarse model grid-boxes (e.g. Schutgens *et al.*, 2016-2017).

*We have now added sections to the main paper (Sect. 3.5.5) and supplementary material (Sect. S2) describing measurement calibration and uncertainty for the instruments used during SAMBBA. We also refer to measurement error in a newly added discussion section (Sect. 3.6 in the revised manuscript) summarising the AOD sensitivity simulations and uncertainty from other sources.*

*We are very familiar with measurement-model sampling uncertainties (e.g. Schutgens et al., 2016; 2017; Reddington et al., 2017) and we have taken care to try and reduce these as much as possible in our analysis. However, we acknowledge that some measurement-model sampling uncertainties will likely remain. We have added the following text to Sect. 3.5.6 (on model spatial resolution):*

"Model spatial resolution will also affect the model–measurement sampling uncertainty, which can be up to 50% for hourly time-resolution data (e.g. Schutgens et al., 2016a; 2017; Reddington et al., 2017). In our analysis we have strived to reduce spatial and temporal sampling errors as much as possible by: 1) running the model and using analysed meteorology for the same time period as the observations; 2) temporally co-locating model and measurement data points, removing time periods with missing or invalid measurement points from the model data (as discussed in Schutgens et al., 2016b) (and temporal averaging for bias calculations and comparisons with aircraft measurements); and 3) spatially co-locating model data to observational data points using interpolation (and spatial averaging for comparisons with aircraft and MODIS observations). For comparisons with aircraft measurements, we have also attempted to reduce measurement representativeness error by removing in-plume and in-cloud sampling from the data where possible. We estimate remaining model–measurement sampling uncertainty to be up to ~30%, corresponding to monthly average model-measurement comparisons (Schutgens et al., 2016a). A higher resolution model would be required to accurately quantify the model–measurement sampling uncertainty for this specific analysis…"

10. Please add the definition of NMBF in the main manuscript and explain how to interpretive it. Would the NMBFs listed in Table S3 benefit from being represented graphically to get a better idea of the model skill? (e.g. Figures 3 in Bender *et al.* 2018).

*We agree that a detailed definition and explanation of the normalised mean bias factor (NMBF) was missing from the submitted manuscript. We have now added this to section 2.3.6 in the revised manuscript.*

*I am unsure whether the reviewer is referring to Table 3 in the original manuscript or Table S1 in the original supplementary material. We agree that a graphical representation of Table S1 (now Table 3 in the revised manuscript) would give the reader a better idea of model skill. We have now included additional figures (Fig. 9, Fig. 10 and Fig. S14) that summarise the model difference and NMBF values for all the sensitivity tests included in the table.*

11. Page 5, L3 – Figures S2 to S5 are referred before Figure S2 (at L15), reorder.

*These figures in the supplementary have now been re-ordered.*

12. Page 5, L29 – Replace Figure 3 by Figure 2. Same at Page6, L9/

*Thank you. Now corrected.*

13. Page 8, L15-17 – "*Sulfate concentrations are well reproduced by the model with no fire emissions and are overestimated when fire emissions are included. This suggests that either emissions of sulfate from fires are overestimated or that other sources of sulfate are overestimated in the model*". I'm struggling to see that from Fig 4b. Is it based on absolute numbers? How does the MACCity compares with say CMIP6 inventories for anthropogenic emissions? It could be nice to show how the different emissions (anthropogenic, dust, BVOCs, …) contribute to the AOT over this region.

*We agree that this is not clear from Fig. 4b, we should have pointed to Table 2 (a reference to Table 2 has now been added).*

*I do not know how the MACCity emissions compare to CMIP6 (CEDs) emissions as there seems to be limited comparisons between aerosol emissions from these inventories in the literature. However, MACCity is based on ACCMIP emissions (developed for CMIP5). MACCity total annual emissions of $SO_2$ from anthropogenic sources are very similar to ACCMIP total annual anthropogenic $SO_2$ emissions over South America for the year 2000 (see figures below; source: https://eccad3.sedoo.fr/). Both MACCity and ACCMIP emissions are widely used in global aerosol models.*

[Figure]

*Calculating and showing how different emissions (anthropogenic, dust, BVOCs, biomass burning) contribute to simulated AOT over this region, would involve multiple sensitivity simulations where each of the emission sources are "switched off"/zeroed out individually. This would be an extensive piece of work and therefore beyond the scope of this study. In the eastern and western Amazon regions selected in our study, biomass burning dominates emissions of $SO_2$, BC and OC over anthropogenic emissions e.g. for total annual emissions:*

| Annual emission (Gg a$^{-1}$) | Western region | | | Eastern region | | |
|---|---|---|---|---|---|---|
| | *OC* | *BC* | *SO$_2$* | *OC* | *BC* | *SO$_2$* |
| *MACCity (2010)* | 6.04 | 2.86 | 12.4 | 14.7 | 7.46 | 30.3 |
| *GFED (2002-2012 mean)* | 546 | 63.3 | 54.4 | 223 | 29.9 | 34.4 |

14. Page 8, L25 – "likely due to a deeper BL over grassland …". Is it confirmed by looking at the model boundary layer height diagnostic? Please define BL acronym also.

    *Thank you – the BL acronym is now defined. We have also now included a figure in the supplementary material (Figure S10) showing the ECMWF ERA-Interim September 2012 mean boundary layer height (calculated from daily means) over the Amazon and wider region. The model meteorology is specified from ECMWF ERA-Interim reanalyses. This figure confirms that the mean boundary layer depth in the eastern region was greater than in the western region. We now refer to this figure in the text (Page 12, line 2 in the revised manuscript).*

15. Figure 3 – Use a different symbol in the legend for 'observations' or a different colour so it stands out from the colour used for 'Model (GFED)'.

    *Done.*

16. Figure 5, L8 – Please state explicitly what STP (i.e. standard temperature & pressure) stands for.

    *Done.*

17. Figure 8, L9 – change purple with grey in the legend.

    *Thank you. Now corrected.*

18. A section on data availability and code availability is necessary to comply with ACP requirements https://www.atmospheric-chemistry-andphysics.net/for_authors/manuscript_preparation.html (see manuscript composition).

    *Thank you. We have now added a section on data and code availability to the revised manuscript.*

**Anonymous Referee #3**

The manuscript by Reddington et al. presents an analysis of biomass burning aerosol abundances over the Amazon during the SAMBBA campaign period using a widely used global aerosol model and a collection of observations in order to assess the performance of the model, but also to provide insight into the processes that contribute to common model biases (primarily the underestimation of biomass burning aerosols). Different state-of-the-art emissions datasets are used for driving the simulations so as to explore the sensitivity of model performance to emissions. A discussion of other potential sources of error is also included. The study is a useful contribution to our understanding of why biomass burning aerosols are currently not well captured in modelling. The manuscript is clearly written and certainly within the remit of Atmospheric Chemistry and Physics. I find the results worthy of publication, following only some minor improvements that I describe below.

*Thank you for the positive comments on our manuscript.*

SPECIFIC COMMENTS:

1. Page 2, Lines 27-33: Worth mentioning here which is the wet and which is the dry season, for non-expert readers. Also, in the last sentence of this paragraph it seems that the brackets need to close.

   *Thank you for the suggestion and correction. Both done.*

2. Page 4, Line 1: "using monthly mean 3-D fields at 6-hourly intervals" – what does this mean?

   *In the version of GLOMAP used in this study, we specify the oxidant concentrations using monthly mean 3-D oxidant fields at 00:00, 06:00, 12:00, and 18:00 each day (so that the diurnal and seasonal variability in oxidant concentrations are both represented).*

3. Page 4, Line 5: What about isoprene?

   *Isoprene SOA is not included in the version of GLOMAP used in this study. We evaluated the GLOMAP-simulated organic carbon (OC) mass concentrations in the wet season against measurements from Manaus (as done in Scott et al. (2018)) and found that the GLOMAP model version used in this study captures the magnitude of measured OC with monoterpene SOA only (see figure below). Which suggests that simulated SOA mass concentrations in this study are reasonable despite not including isoprene SOA. We note the new analysis of McFiggins et al. (2019) showing no net SOA production from isoprene.*

4. Page 4, Line 7: "Size-resolved emissions of mineral dust are prescribed from daily varying emissions fluxes" – Not clear what this means. Are they dependent on the model's meteorology?

   *Size-resolved emissions of mineral dust in GLOMAP are prescribed from daily varying emissions fluxes for the year 2000 provided for AEROCOM (generated by the NASA Goddard Earth Observing System Data Assimilation System) (Dentener et al., 2006). Thus the dust emissions depend on meteorology in the year 2000; not the GLOMAP model meteorology specified by ECMWF ERA-Interim analyses for 2012.*

5. Page 4, Lines 21-25: It is worth giving some additional (brief) information on the fire emissions datasets, e.g. on how they are produced (e.g. based on burnt area or fire radiatve power etc.). Also, any ideas from the literature on why they differ the way they do over different parts of the Amazon (as discussed in the subsequent paragraphs)?

   *We have now included a new section describing the fire emissions datasets and including information on updates that have been documented (please see Sect. 2.2.2 in the revised manuscript).*

6. Page 7, Line 13: I realised after finishing reading Section 2 that the start and end date of the simulation as well as the spin-up period have not been mentioned.

   *Thank you for pointing this out. We have now added an addition section (Sect. 2.1.1) describing the model simulation length, output frequency and spin-up time.*

7. Page 7, Lines 22-23: I wonder what it is that made all of them miss the fire emissions just for that part of the season and not for the rest of it. Is it not possible that this could be due to the atmospheric modelling? For example due to the meteorological conditions not being captured well?

   *We agree that the model meteorology might be partly responsible for the failure of the model to capture the decrease in aerosol concentrations from Phase 1 to Phase 2 (particularly in the vertical profile of aerosol mass). We have added the following text to the revised manuscript in Sect 3.2:*
   "This may be because the emission datasets report only moderately lower emissions in Phase 2 compared to Phase 1 (Figure S3a; Table 1), but also because the model may underestimate wet removal of aerosol during Phase 2 (consistent with model and observation comparisons in Archer-Nicholls et al. 2015)."

8. Page 8, Line 17: "or that other sources of sulfate are overestimated in the model" – the fact that before the fire season there are no overestimations probably suggests that non-fire emissions are not responsible?

   *We unfortunately do not have measurements of sulphate aerosol mass outside the SAMBBA campaign period (and fire season) in 2012 to test this. We have total PM2.5 mass concentration measurements, which as the referee points out, the simulated PM2.5 concentrations agree well with during the wet season. However, as sulphate is likely to remain a fairly minor component of PM2.5 mass concentration (with organic matter likely dominating), we cannot deduce anything about model sulphate outside the fire season.*

9. Page 8, Line 25: "and is likely due to a deeper BL over grassland vegetation in the eastern Amazon" – any explanation or reference to support this?

*We have now included a figure in the supplementary material (Figure S10) showing the ECMWF ERA-Interim September 2012 mean boundary layer height (calculated from daily means) over the Amazon and wider region. This figure confirms that the mean boundary layer depth in the eastern region was greater than in the western region in September 2012. We now refer to this figure in the text (Page 12, line 2 in the revised manuscript).*

10. Page 9, Line 12: "in during" -> "during".

*Thank you. Now corrected.*

11. Page 11, Line 7: The biases are, however, smaller when comparing with AERONET measurements. Worth mentioning, and perhaps commenting on. What is the uncertainty in observations themselves?

*We have now adjusted the size of the region in the west that we average the MODIS data over, to be more in-line with Darbyshire et al. (2018) and to be more consistent with the region impacted by deforestation fires (in the previous version of the manuscript we were averaging MODIS and model AOD550 over large sections of pristine forest). The model biases in AOD500 (against AERONET) and AOD550 (against MODIS) are now more consistent:*

|  | noBBA | FINN | GFED | GFAS |
|---|---|---|---|---|
| Western Amazon, Phase 1 | | | | |
| AOD550 (MODIS) | -5.25 | **-0.51** | -0.98 | -1.43 |
| AOD500 (AERONET) | -6.95 | **-0.47** | -0.53 | -1.26 |
| Western Amazon, Phase 2 | | | | |
| AOD550 (MODIS) | -3.68 | **-0.38** | -0.70 | -1.06 |
| AOD500 (AERONET) | -4.50 | **-0.41** | -0.68 | -1.11 |

*If we reduce the averaging region for MODIS AOD further (to fit more tightly around the western AERONET stations: 54-68.5°W, 7-12°S) and compare model biases in AOD550 over this region to model biases in AOD500 only at the three AERONET stations in the western region, again we see that they are consistent (e.g., with GFED emissions: AERONET AOD500, $NMBF_{P1}$=-0.58, $NMBF_{P2}$=-0.49; MODIS AOD550, $NMBF_{P1}$=-0.64, $NMBF_{P2}$=-0.44).*

*We have added a section to the supplementary material describing further details of the instrumentation used during SAMBBA; including information about measurement calibration and uncertainty (please see new Sect. S2). For AOD retrievals, the 1σ uncertainty is estimated to be ~0.05±0.15% for MODIS and ~0.01 AERONET.*

*We have made the following changes and additions to the text in Sect. 3.4:*

"Consistent with comparisons to MODIS, the model generally underestimates AOD500 at all stations and with all fire emissions, except at two stations in the western Amazon (Rio Branco (in both campaign phases) and Porto Velho (in Phase 2)) with FINN emissions. The negative model bias in AOD500 across all AERONET stations is consistent with the negative model bias in AOD550 (against MODIS) (Table 2), but is smaller at some individual stations (Fig. 8). This is likely due to multiple reasons including differences in: i) the AOD wavelengths (500 nm versus 550 nm); ii) the AERONET and MODIS retrieval uncertainties (Sect. S2.3); iii) the

location/region of comparison, affecting magnitude and sources of AOD; and iv) the AERONET and MODIS data coverages."

12. Sect 3.5 (general): Given that this section is a substantial fraction of the manuscript, it would have been nice to support it with some figure or table in the main text. Maybe one simple thing to do is move Table S1 to the main part of the manuscript.

    *Table S1 has now been moved into the main manuscript (as Table 3) and extended to include additional sensitivity tests. We have also included additional figures (Fig. 9, Fig. 10, Fig. S13, and Fig. S14) to support Sect. 3.5 and Table 3.*

13. Page 12, Lines 28-29: Size distribution and composition are not discussed substantially in this analysis of possible factors. Is there anything more that could be said about them? Also, what about meteorological quantities other than humidity, e.g. wind or temperature?

    *We have now included an additional section discussion size distribution in relation to simulated AOD (Sect. 3.5.1 in the revised manuscript). Other meteorological factors may also play a small role e.g. temperature and wind (although this would also affect simulated mass concentrations). However, exploring their contribution and evaluating these quantities against observations would be an extensive piece of work and beyond the scope of the current study.*

14. Page 13, Lines 18-19: "but not fully resolve the negative bias in model AOD" – please add "...which is of the order of XX%" to give a sense of how far the results would still be.

    *We have now altered this sentence in this section and instead added a new section (Sect. 3.6 in the revised manuscript) summarising the AOD sensitivity simulations and discussing the various uncertainties associated with the AOD calculation and other sources.*

**References used in the author response**

[revised manuscript text omitted]